# M1BP is an essential transcriptional activator of oxidative metabolism during *Drosophila* development

Gabriela Poliacikova [1,2], Marine Barthez[1,2], Thomas Rival [1], Aïcha Aouane[1], Nuno Miguel Luis [1], Fabrice Richard[1], Fabrice Daian [1], Nicolas Brouilly[1], Frank Schnorrer [1], Corinne Maurel-Zaffran [1], Yacine Graba[1] & Andrew J. Saurin [1] ✉

Oxidative metabolism is the predominant energy source for aerobic muscle contraction in adult animals. How the cellular and molecular components that support aerobic muscle physiology are put in place during development through their transcriptional regulation is not well understood. Using the *Drosophila* flight muscle model, we show that the formation of mitochondria cristae harbouring the respiratory chain is concomitant with a large-scale transcriptional upregulation of genes linked with oxidative phosphorylation (OXPHOS) during specific stages of flight muscle development. We further demonstrate using high-resolution imaging, transcriptomic and biochemical analyses that Motif-1-binding protein (M1BP) transcriptionally regulates the expression of genes encoding critical components for OXPHOS complex assembly and integrity. In the absence of M1BP function, the quantity of assembled mitochondrial respiratory complexes is reduced and OXPHOS proteins aggregate in the mitochondrial matrix, triggering a strong protein quality control response. This results in isolation of the aggregate from the rest of the matrix by multiple layers of the inner mitochondrial membrane, representing a previously undocumented mitochondrial stress response mechanism. Together, this study provides mechanistic insight into the transcriptional regulation of oxidative metabolism during *Drosophila* development and identifies M1BP as a critical player in this process.

Mitochondria, the energetic hubs of the cell, are best known for harbouring pathways required for efficient ATP synthesis through the tricarboxylic acid (TCA) cycle and oxidative phosphorylation (OXPHOS). These essential, evolutionary conserved organelles are at the core of cellular metabolism[1] and as such their dysfunction lies central to a plethora of chronic human diseases, including myopathies, multiple encephalopathies and neuropathies, diabetes, epilepsy and cancer (reviewed in ref. [2]).

The OXPHOS system consists of the respiratory chain complexes I to IV and the mitochondrial $F_1F_o$-ATP synthase, herein referred to as Complex V. Mitochondrial content and the expression of genes encoding the OXPHOS complex system are highly dynamic to adapt to the metabolic and energetic demands of the different tissues of an organism[3]. The timely transcriptional control of nuclear genes encoding proteins of various metabolic pathways is essential for proper mitochondrial biogenesis and function, since mitochondria

[1]Aix-Marseille Univ, CNRS, Developmental Biology Institute of Marseille (IBDM), UMR 7288, Case 907, Turing Center for Living Systems, Parc Scientifique de Luminy, 13288 Marseille Cedex 09, France. [2]These authors contributed equally: Gabriela Poliacikova, Marine Barthez. ✉e-mail: andrew.saurin@univ-amu.fr

import the vast majority of their proteins from the cytoplasm[4–6]. Numerous transcription factors and transcriptional co-activators have been identified that regulate the transcription of nuclear-encoded mitochondrial genes implicated in mitochondrial function, biogenesis and OXPHOS (herein referred to as mito-genes[7]) (reviewed in refs. 8–10). The role these factors play in mitochondrial adaptation to environmental cues, stresses or disease has been extensively studied, yet how this is coordinated in a developmental context is less well known.

Muscles are one of the most energetically demanding tissues in the body, requiring large amounts of ATP for their contraction and proper function[11]. Several fibre types can be distinguished according to their contractile and metabolic properties. Slow-twitch, type I fibres exhibit high resistance to fatigue, are enriched in postural muscles and are largely oxidative. Fast-twitch, type II fibres, are quickly fatigable with high power outputs and are enriched in directional muscles. With a decreasing oxidative capacity, we can further distinguish fast-oxidative (IIA), fast-intermediate (IIX) and fast-glycolytic (IIB) fibres (reviewed in ref. 12). Mitochondria dynamics, orientation and content vary between different fibre types, with oxidative fibres displaying higher individual mitochondrial volume, a grid-like mitochondrial network and higher fusion rates, while mitochondria in glycolytic fibres show a perpendicular orientation to the contraction axis and less connectivity[13–15].

Studies conducted in vitro and in postnatal muscles have demonstrated that proteins associated with mitochondrial biogenesis and respiration increase sharply during muscle differentiation[16–19]. Even though several factors regulating mitochondrial metabolism have been identified, their requirement at different stages of muscle development and between different muscle fibre types are poorly understood.

In *Drosophila*, adult muscle development has been extensively studied, the most prominent adult muscles being the highly oxidative indirect flight muscles (IFMs) oscillating at 200 Hz during flight[20]. A second muscle type is present in the adult legs or abdomen mediating locomotion or mating. Remarkably, the mitochondrial morphology between both muscle types is strikingly different with flight muscle mitochondria containing very dense cristae and being in very close contact with the contractile myofibrils, in contrast to the more complex shapes of leg muscle mitochondria that locate more sparsely in the middle and at the periphery of myofibrils[21].

Indirect flight muscles are formed from myoblasts that are attached to the notum region of the larval wing disc. During early pupation, myoblasts migrate and fuse with surviving larval muscle scaffolds, the larval oblique muscles (LOMs), to form the dorsal longitudinal flight muscles (DLMs). In contrast, dorso-ventral flight muscles (DVMs) form de novo, without any larval scaffolds. DLMs and DVMs form collectively the adult indirect flight muscles[22,23], that we will for simplicity herein refer to as flight muscles. Since mitochondrial flight muscle development takes place during the accessible pupal stage[22], *Drosophila* is an excellent model to study the molecular control of mitochondrial biogenesis and dynamics during muscle development.

Current data suggest that several transcription factors and coactivators are involved in mitochondrial oxidative metabolism control during flight muscle development. The transcription factor Spalt-major (encoded by the *salm* gene), known for its role in muscle fibre identity[24] has been shown to regulate the expression of both sarcomeric and mitochondrial genes during myogenesis[7,21]. Specifically, Spalt is involved in the upregulation of TCA and OXPHOS protein coding genes after 30 h after puparium formation (APF) in flight muscles, although it is unclear whether it is directly or indirectly activating the expression of these genes[7,21]. Retinoblastoma-family protein (Rbf) has been shown to directly activate the transcription of both sarcomeric and mitochondrial respiratory chain-encoding genes, but the functional consequences on mitochondria development upon Rbf loss of function were not studied[25].

Interestingly, it was shown that a motif enriched in genes activated by Rbf is a motif bound by Motif-1 binding protein (M1BP)[25,26]. M1BP is an RNA polymerase II pausing factor, binding to the promoters of genes involved in basic cellular processes such as cell cycle and metabolism. We have previously shown that M1BP plays an important role in the control of metabolism since one-quarter of all metabolic genes are downregulated upon fat body-specific M1BP knock-down[27]. These data prompted us thus to ask whether M1BP function in metabolism control is important in *Drosophila* muscles.

In this study, we performed an ultrastructural imaging description and detailed transcriptional and biochemical analyses of mitochondrial oxidative metabolism during the course of *Drosophila* flight muscle development. This work identifies an essential role for M1BP in activating the transcription of nuclear-encoded OXPHOS-related genes and in controlling mitochondrial respiratory chain assembly. This approach provides a detailed analysis of *Drosophila* flight muscle mitochondrial development and identifies a critical player in the regulation of muscle mitochondrial oxidative metabolism.

## Results
### M1BP loss leads to ultrastructural defects in mitochondria
It had been shown that OXPHOS genes activated by Rbf in flight muscles contain Motif-1[25], which is bound by Motif-1 binding protein[26]. M1BP is a transcription factor that regulates the transcription of metabolic genes in the larval fat body[27] and in in vitro cell culture systems[26,28]. Based on these observations, we wished to determine whether M1BP is involved in the regulation of mitochondrial metabolism in the flight muscles.

Throughout flight muscle development, transcriptome data demonstrate that *M1BP* expression is relatively stable (data analysed from[7] and represented in the Supplementary Fig. 1a), which we confirmed by immunolabelling in both wing disc-associated myoblasts (labelled by a *twi*::*GFP* transgene[29]) and adult flight muscle nuclei (Supplementary Fig. 1b, c). Driving UAS-M1BP RNAi construct by the muscle-specific *Mef2-GAL4* driver (Mef2 > *M1BP*-RNAi), expressed during all myogenic stages[30], we knocked down *M1BP* expression by ≈40–60% in myoblasts and in the adult muscle (Supplementary Fig. 1b', c', c''). It has been previously reported that expressing *M1BP* RNAi in in vitro cell culture conditions leads to the eventual cell cycle arrest following few days of RNAi expression[26,31]. We thus wanted to test whether M1BP knockdown (KD) in myoblasts will affect their proliferation. To count the number of wing disc-associated nuclei, we developed a machine learning-based, 3D nuclei counting technique that allowed us to estimate the number of myoblasts on the wing disc notum region at 0 h APF to approximately 1500 cells. As we did not detect any significant decrease in myoblast number upon *M1BP* RNAi driven by *Mef2-GAL4*, we thus conclude that M1BP does not regulate myoblast proliferation (Supplementary Fig. 1d).

To test whether M1BP plays a role in the regulation of flight muscle metabolism, we first assessed the muscle function after M1BP down-regulation by a standard flight assay[32]. Flight muscles depleted of M1BP during development do not support flight, with 50–100% of adult males (depending on the RNAi line used), being flightless (Fig. 1a). For *M1BP* RNAi #1 we further showed that even though a milder percentage of adults are flightless, the flight muscles in this condition are less resistant to fatigue compared to the WT (Supplementary Fig. 2a). To identify the origin of these phenotypes, we performed an immunolabelling of adult DLMs. *M1BP* RNAi driven by *Mef2-GAL4* results in no visible perturbation of adult sarcomeric integrity and size (Fig. 1b, b'). Mitochondria in *M1BP*-depleted muscles, stained with an anti-Complex V antibody (ATP5A subunit, encoded by the *bellwether* gene), are present in between myofibrils, although they display a severe defect in their shape (Fig. 1b, bottom panels) that prompted us to investigate the mitochondria further.

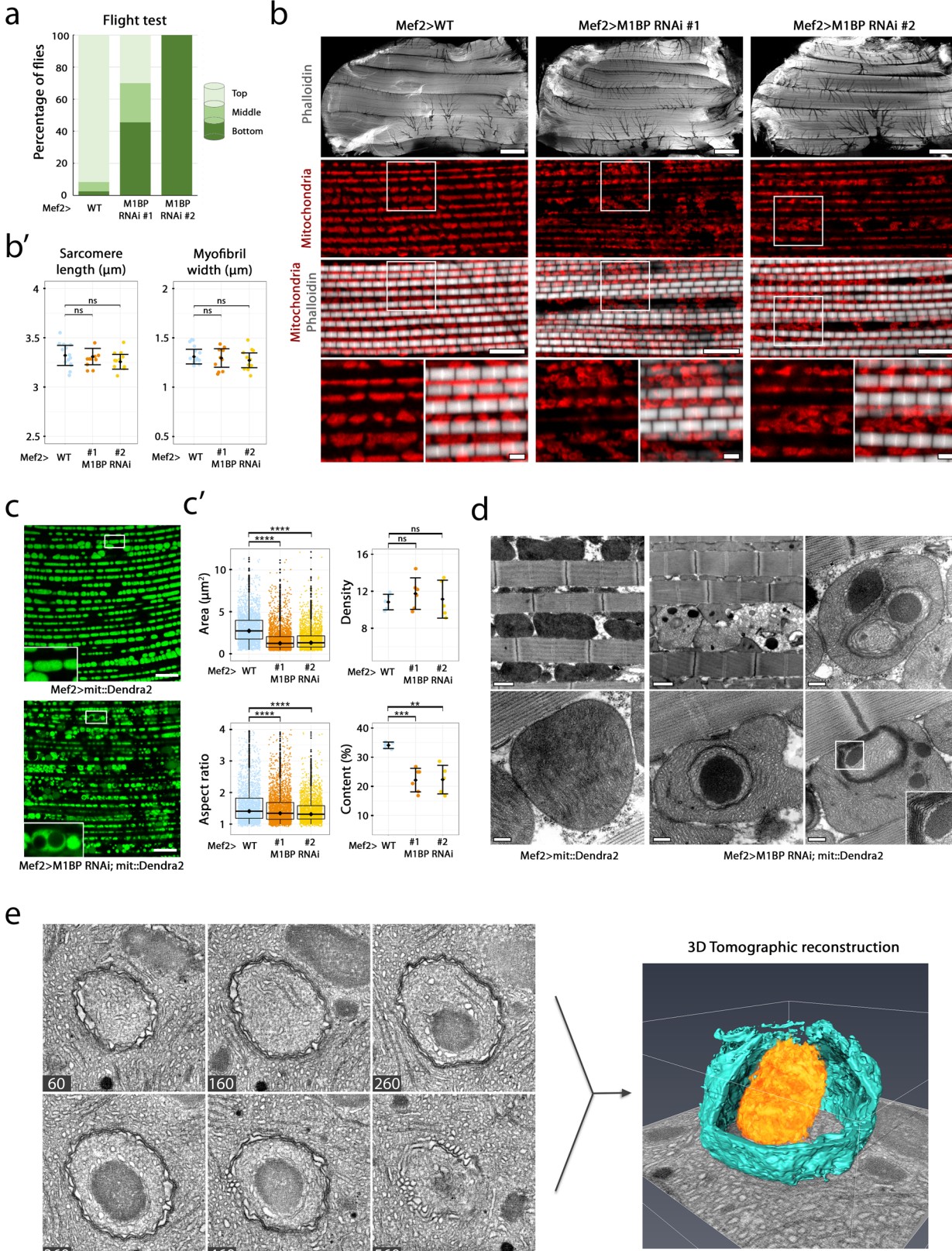

Knowing that the *Mef2-GAL4* transgene can drive expression in a subset of neurons[33] and that *Mef2* is expressed also in the fat body[34], we wished to verify that the muscle mitochondrial phenotypes we observed upon *M1BP* KD are not due to perturbation of *M1BP* expression in these other tissues. This is particularly important for the fat body, given the known intercommunication between fat body and

muscle[35–37] and that transcription factor perturbation in one can result in a phenotype in the other[38]. To this end, we downregulated *M1BP* with either a pan-neural *elav-GAL4* driver or a fat body-specific *cg-GAL4* driver, yet observed no sarcomeric or mitochondrial phenotypes in the flight muscles (Supplementary Fig. 2b, c). Furthermore, we confirmed the muscle paralysis and mitochondrial morphology defects upon

**Fig. 1 | M1BP downregulation in flight muscles leads to mitochondrial ultra-structural defaults. a** Flight ability was scored as a mean percentage of male flies landing on column segments, from two independent biological replicates ($n_{WT} = 150$, $n_{\#1} = 90$ and $n_{\#2} = 20$). **b** Confocal sections of female wild-type (left) and M1BP-depleted DLMs with two different RNAi lines, (#1 and #2) driven by *Mef2-GAL4*. White rectangle represents an enlarged view. Myofibrils are stained with phalloidin (grey) and mitochondria with anti-ATP5A antibody (red). The scale bar is 100 μm (top), 10 μm (middle) and 2 μm (bottom views). **b'** Quantification of muscle parameters, each dot corresponding to one animal ($n_{WT} = 15$, $n_{\#1} = 10$ and $n_{\#2} = 11$ flies). Mean and standard errors are indicated. After applying Shapiro–Wilk test for normality, two-sided Mann–Whitney test was applied (p-values (length) = 0.87 and 0.1 and p-values (width) = 0.53 and 0.44). **c** Confocal sections of live 2-days-old female adult wild-type (left) and M1BP-depleted (right) DLM mitochondria labelled with mit::Dendra2, driven by *Mef2-GAL4*. Scale bars represent 10 μm. **c'** For area and aspect ratio quantification, data are represented in boxplots (median and quartiles) with whiskers (minimum to maximum), where each point represents one mito-chondrion ($n_{WT} = 2464$, $n_{\#1} = 3196$ and $n_{\#2} = 2535$ mitochondria). Outliers are depicted with black circles. Two-sided Mann–Whitney test was applied (p values (area) = $2 \times 10^{-16}$ **** and (aspect ratio) = $2.9 \times 10^{-9}$ and $2.8 \times 10^{-9}$ ****). For density and content quantification mean and standard errors are indicated, where each point represents one confocal section ($n_{WT} = 5$, $n_{\#1} = 6$ and $n_{\#2} = 6$ sections). After applying Shapiro–Wilk test for normality an unpaired two-sided Student's test was applied (p values (density) = 0.28 and 0.76 and (content) = 0.0005 *** and 0.005 ***). **d** TEM micrographs of male adult WT (left) and *M1BP*-depleted (middle, right) DLM mitochondria. Scale bars represent 1 μm (larger views) and 200 nm (individual mitochondria). **e** Tomographic reconstruction of an inclusion (orange) and the IMM (blue) (to visualise the inclusion a part of the IMM was computa-tionally removed). Six of the 633 slices from the reconstructed tomogram stack are shown with their respective stack position. Source data are provided as a Source data file.

*M1BP* KD with another myoblast-specific driver, *Him-GAL4*, that in our hands downregulates *M1BP* expression until the adult stage to a similar level as the *Mef2-GAL4* driver (Supplementary Fig. 2d, e), demonstrat-ing that the phenotypes we observe in flight muscle are due to M1BP function in this tissue.

To better characterise the mitochondrial morphology defects we observed, we used a mitochondrially localised fluorescent protein, mit::Dendra2, enabling live mitochondrial imaging in *Drosophila* tissues[39]. While wild-type mitochondria display the well described regular ellipsoid shapes[21], *M1BP*-depleted mitochondria are severely disorganised and display heterogeneous fluorescent intensity (Fig. 1c). We quantified a number of mitochondrial parameters: area, aspect ratio (ratio of major to the minor axis), density (number of mito-chondria per 100 μm²) and mitochondrial content (mitochondrial area with respect to total area), on live mitochondria to avoid fixation artefacts (Fig. 1c'). We observed that mitochondria in muscles from *M1BP* RNAi are smaller than WT mitochondria (the area median of 2.8 μm² in WT compared to 1.29 μm² and 1.36 μm² in two distinct *M1BP* RNAi expressing lines) and slightly rounder (WT median aspect ratio of 1.4 compared to 1.3 in both *M1BP* RNAi-expressing lines). Furthermore, we detected a significantly lower mitochondrial content upon *M1BP* KD (mean of 34% in WT and 22% in both *M1BP* RNAi-expressing lines), with no significant change in the mitochondrial density (mean of 11 mito-chondria per 100 μm² in WT vs 11.7 and 11.1 in *M1BP* KD). Together, this suggests that mitochondria are smaller due to a decrease in their biogenesis rather than an increase in their fission. Importantly, we observed many mitochondria containing large circular zones devoid of mit::Dendra2 staining (see Fig. 1c inset).

Transmission electron microscopy (TEM) confirmed that the adult sarcomere ultrastructure is unperturbed by M1BP RNAi, as assayed by longitudinal (Fig. 1d) and cross-sections (Supplementary Fig. 3a) of adult flight muscles. While wild-type adult flight muscle mitochondria display dense, parallel, lamellar cristae spanning the mitochondria, *M1BP* downregulation leads to widespread mitochon-drial ultrastructural defects (Fig. 1d): while we observed several instances of translucid mitochondria that are likely being eliminated (Fig. 1d, top middle panel), the most prominent, widespread defects were large, amorphous, electron-dense inclusions in the mitochondrial matrix. We noticed that in adult mitochondria, these inclusions were often encircled by several rounds of inner mitochondrial membrane (IMM), with the mitochondrial matrix enclosed within being more electron-lucent (Fig. 1d, middle bottom and right panels). We hypo-thesise that the inclusions either represent a spatial constraint on developing cristae that end up by encircling them during cristae development or that mitochondria actively attempt to isolate the inclusions from the remaining mitochondrial matrix, which could account for the more electron-lucent matrix surrounding the inclu-sions and correspond to the observed mit::Dendra2-excluded zones in live imaging. To test whether IMM encirclement could serve for compartmentalisation of inclusions we performed serial electron tomography 3D-imaging. The reconstruction showed that the inner mitochondrial membrane completely encompasses the electron-dense structures, thereby isolating it from the rest of the matrix (Fig. 1e and Supplementary Movie 1), which is suggestive of an active mitochondrial process to isolate the inclusion using the IMM that has not yet been described.

We confirmed the presence of mitochondrial inclusions in the flight muscles upon *M1BP* KD with another muscle-specific driver, *Him-GAL4* (Supplementary Fig. 3b). Furthermore, mitochondrial inclusions can also be observed upon *M1BP* KD in the tubular leg muscles upon *Mef2-Gal4* KD (Supplementary Fig. 3c) and in the highly metabolically active fat body upon *cg-GAL4* KD (Supplementary Fig. 3d). Of note here, like for the flight muscle mitochondrial inclusions that result in flight defects, leg muscle function is severely perturbed, which we attribute to the mitochondrial defects observed since *M1BP* KD results in flies that are unable to climb and are severely restricted in the speed at which they walk (Supplementary Fig. 3c'). This mitochondrial phe-notype thus includes both fibrillar and tubular muscle types and even spans to other metabolically active tissues.

Overall, these data demonstrate that M1BP downregulation in the muscle leads to widespread adult mitochondrial morphological defects, ultimately resulting in muscle paralysis as observed through a flightless and climbing phenotype.

## OXPHOS transcription is concomitant with cristae biogenesis

Having shown that flight muscle mitochondria upon *M1BP* KD con-tain large inclusions in their matrix, we wished to determine the developmental origin of this phenotype. It is widely accepted that mitochondria undergo regulated changes during the muscle differ-entiation process[13,17]. Several studies imaged mitochondrial dynamics during flight muscle development using fluorescence microscopy that does not allow for proper cristae visualisation[21,40,41], while a study focusing on the trachea invasion of flight muscles used electron microscopy imaging but was limited to two developmental time points[42].

Given the little available ultra-structural data about mitochondrial development during flight myogenesis, we first decided to better describe this process in the wild-type situation, and second char-acterise the function that M1BP plays. To characterise mitochondrial morphological ultrastructural changes during flight myogenesis in more detail, we used high-resolution, transmission-electron micro-scopy. We performed a time-course of flight muscle mitochondrial development spanning from third instar larval myoblasts until the adult stage (Fig. 2a and Supplementary Fig. 4a). Wing disc-associated myoblasts give rise to flight muscles during pupation[22]. Their mito-chondria at 0 h APF have few cristae (mean of 6.9 cristae per 500 nm), which does not change during early stages of myogenesis through to 48 h APF (mean density of 6.3 cristae; quantification of cristae at 34 h

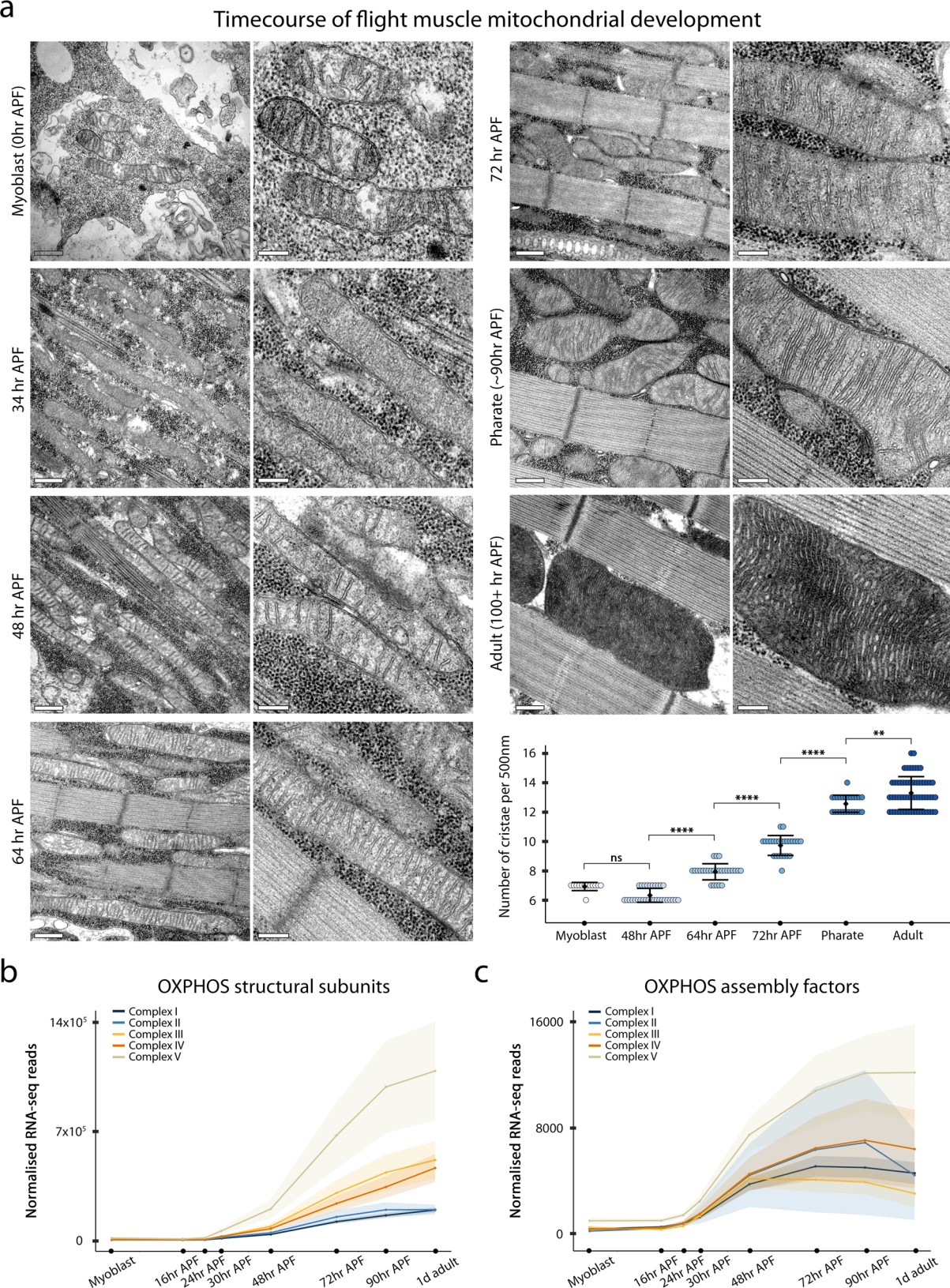

**a** Timecourse of flight muscle mitochondrial development

**b** OXPHOS structural subunits

**c** OXPHOS assembly factors

APF was not performed as they were poorly defined by TEM). The density of cristae starts to increase around 64 h APF (mean density of 7.9 cristae) and continues increasing throughout development to eclosion (mean density of 13.3 in 2-day-old adults) (Fig. 2a). The number of cristae appears to peak during the first week of adult life (mean density of 15.2 in 7 days-old adults), since the number

of cristae in aged flies decreases to the values observed at eclosion (mean density of 13.3 in 30-day-old adults) (Supplementary Fig. 4b).

During the early stages of myogenesis, mitochondria intercalate between immature myofibrils and elongate into large tubules[21,41]. While cristae biogenesis has not yet started, we observed a sharp decrease in the mitochondrial circularity and increase in perimeter between 24 h

**Fig. 2 | Large scale OXPHOS-related gene transcriptional upregulation and mitochondrial cristae biogenesis occur during flight muscle development. a** Transmission electron microscopy micrographs of wild-type flight muscle development spanning from wing-disc associated myoblasts to 2 days-old adult DLM. For each time point, a larger view (left, scale bar of 500 nm) and a zoom on mitochondria cristae (right, scale bar of 200 nm) are represented. The quantification of the number of mitochondrial cristae per length unit (500 nm) is shown for each corresponding time point except for 34 h APF. Mean and standard deviation are shown with each point representing a value of a single mitochondrion (*n* (Myo) = 12, *n* (48 h) = 19, *n* (64 h) = 15, *n* (72 h) = 23, *n* (pharate) = 21 and *n* (adult) = 47 mitochondria). For statistical analysis, one-way analysis of variance (ANOVA) was performed, followed by Tukey's test for multiple mean comparisons (*p* values: Myo vs 48 h = 0.2; 48 h vs 64 h = 0 ****; 64 h vs 72 h = 0 ****; 72 h vs pharate = 0 ****; pharate vs adult = 0.002 **). **b, c** Transcriptional profile during flight muscle development of genes encoding structural subunits (**b**) and assembly factors (**c**) of the mitochondrial respiratory chain complexes. The solid lines represent the normalised mean of sequenced counts of all subunits/assembly factors of a given respiratory complex with shaded areas representing the standard error of the mean. Source data are provided as a Source data file.

and 34 h APF, which might be important for proper myofibril development (Supplementary Fig. 4c)[21]. Of note is that between the pharate and the adult stages, mitochondria increase significantly their area and perimeter to adopt their typical ellipsoid shape and squeeze densely into a single row in between adult myofibers[21] (Supplementary Fig. 4c).

Mitochondrial cristae bear the OXPHOS protein complex chain, comprised of five multimeric complexes that are comprised of numerous core enzymatic subunits, accessory subunits and proteins required for their assembly (reviewed in ref. [43]). To complement our imaging analysis and provide further insight into mitochondrial metabolic states during flight muscle development, we performed a computational analysis of OXPHOS gene expression during flight myogenesis using available transcriptome data performed at similar developmental time points[7]. Using Flybase[44] and Gene Ontology (GO)[45] databases we manually curated a list of *Drosophila* nuclear genes encoding core and accessory subunits (herein called structural subunits; 73 genes) as well as genes encoding proteins required for complex assembly (termed assembly factors; 52 genes) of the *Drosophila* mitochondrial respiratory chain (that we will collectively refer to in this article as OXPHOS genes) (Supplementary Data 1). We observed that OXPHOS gene expression has two phases: first a basal expression phase that occurs from third instar larval myoblasts until 48 h APF (after puparium formation), followed by a large-scale transcriptional upregulation phase sustained until the adult stage (Fig. 2b, c and Supplementary Data 1). The transcriptional upregulation of these genes is complex-dependent: genes encoding subunits of Complex V are transcribed at the highest level, Complexes III and IV subunits have a similar transcriptional profile with a number of reads being approximately half than those of Complex V, and Complexes I and II have the lowest transcription levels, being approximately 20% of those of Complex V (Fig. 2b).

The transcriptional upregulation of OXPHOS assembly factors slightly precedes the upregulation of structural subunits and peaks between 70–90 h APF before showing trends of downregulation at the adult stage (Fig. 2c and Supplementary Fig. 5a). Similar to the structural subunits, the transcripts of genes encoding assembly factors of Complex V are the most abundant. Assembly factors of other complexes have similar expression profiles with approximately half the number of transcripts compared to Complex V.

Continuing our analyses of the developmental transcriptome data from Spletter and colleagues[7], we extended this to genes coding for other critical OXPHOS components: creatine kinases (in *Drosophila* arginine kinases), involved in ATP buffering[46], adenosine nucleotide translocases (ANTs), mediating ADP/ATP transport[47] and the mitochondrial phosphate carrier (PiC), involved in the inorganic phosphate transport across the IMM[48]. The expression of *Drosophila* orthologue of vertebrate creatine kinases, *Arginine kinase 1* (*Argk1*)[49] peaks at the mid-myogenesis (Supplementary Fig. 5b). Concerning *Drosophila* orthologues of vertebrate ANTs, *Ant1*, also called *stress sensitive B* (*sesB*) and *Ant2*[50], only *Ant1* transcripts were detected during flight myogenesis and its expression dynamics follows those of OXPHOS subunits and assembly factors (Supplementary Fig. 5c). Similarly, for the *Drosophila* orthologues of PiC, *Mitochondrial phosphate carrier proteins 1* (*Mpcp1*) and 2 (*Mpcp2*)[51], only *Mpcp1* expression was

detected and its expression follows similar dynamics to *Ant1* and OXPHOS-related genes (Supplementary Fig. 5d).

The expression of genes involved in the TCA cycle and glycolysis follows very similar dynamics to OXPHOS genes, which is much less the case for genes involved in other mitochondrial processes such as amino acid metabolism or the pentose phosphate pathway (Supplementary Fig. 5e). Together, these data highlight a large-scale transcriptional response of genes required for mitochondrial energy metabolism that correlate with extensive cristae biogenesis.

## M1BP regulates OXPHOS gene expression during myogenesis

The *GAL4/UAS* system is a temperature-sensitive system, with increased temperature driving more *UAS-transgene* expression[52]. We noticed that rearing flies expressing *Mef2-GAL4* driven *M1BP* RNAi at 29 °C was lethal at the pharate stage, whereas at 25 °C only a third of flies eclose likely due to an incapacity of the fly to break the pupal case required for eclosion. We thus used this lethality phenotype to determine when M1BP function is required during adult myogenesis. Temporal control of *M1BP* RNAi expression during myogenesis using the TARGET system, in which GAL4 is active at 29 °C and inactive at 18 °C[53], demonstrated an essential role for M1BP during early-to-mid pupation, since down-regulation of *M1BP* between ≈10–60 h APF leads to the pharate pupal lethality phenotype (Fig. 3a). This fits with stages at which the transcription of components involved in mitochondrial energy metabolism is increasing (see above).

To study the origin of mitochondrial defects in M1BP-depleted flight muscles we imaged mitochondria at three different time points, representative of distinct mitochondrial metabolic states we established earlier: myoblasts, in which the mitochondrial respiratory chain is poorly developed; 48 h APF, at which stage mitochondria are at the peak of their elongation and OXPHOS genes start to be transcriptionally upregulated; and 64 h APF when the process of cristae biogenesis is fully engaged (see Fig. 2 and Supplementary Figs. 4 and 5). We did not detect any major morphological changes in mitochondria in larval myoblasts or at 48 h APF upon *Mef2-GAL4* driven *M1BP* RNAi (Fig. 3b). In contrast, at 64 h APF, we observed that mitochondria in M1BP-depleted muscle display fewer cristae (mean density of 7.9 cristae in WT versus 6 upon *M1BP* KD) and contain small electron-dense, amorphous inclusions in their matrix, similar to those observed in adult *M1BP* KD mitochondria (Fig. 3b, bottom panels). We thus conclude that mitochondrial inclusions observed upon *M1BP* KD are of developmental origin, appearing around 60 h APF, at a time when we first observed an increase in the number of cristae in wild-type mitochondria (see Fig. 2).

Mitochondrial inclusions have been reported in patients suffering from myopathies[54] and cancer[55] but their origin has not been determined. Nevertheless, mitochondrial inclusions seem to be related to the mitochondrial respiratory chain defects since they have also been observed in mitochondria of rats fed with the respiratory toxin (cronotoxin)[56] and in yeast mutants for subunits of the Complex V[57]. We thus looked at transcriptomic changes in genes with mitochondrial function triggered by M1BP depletion in flight muscles that we reasoned could give us insight into the origin of mitochondrial inclusions we observed. We performed RNA-seq analyses at the same

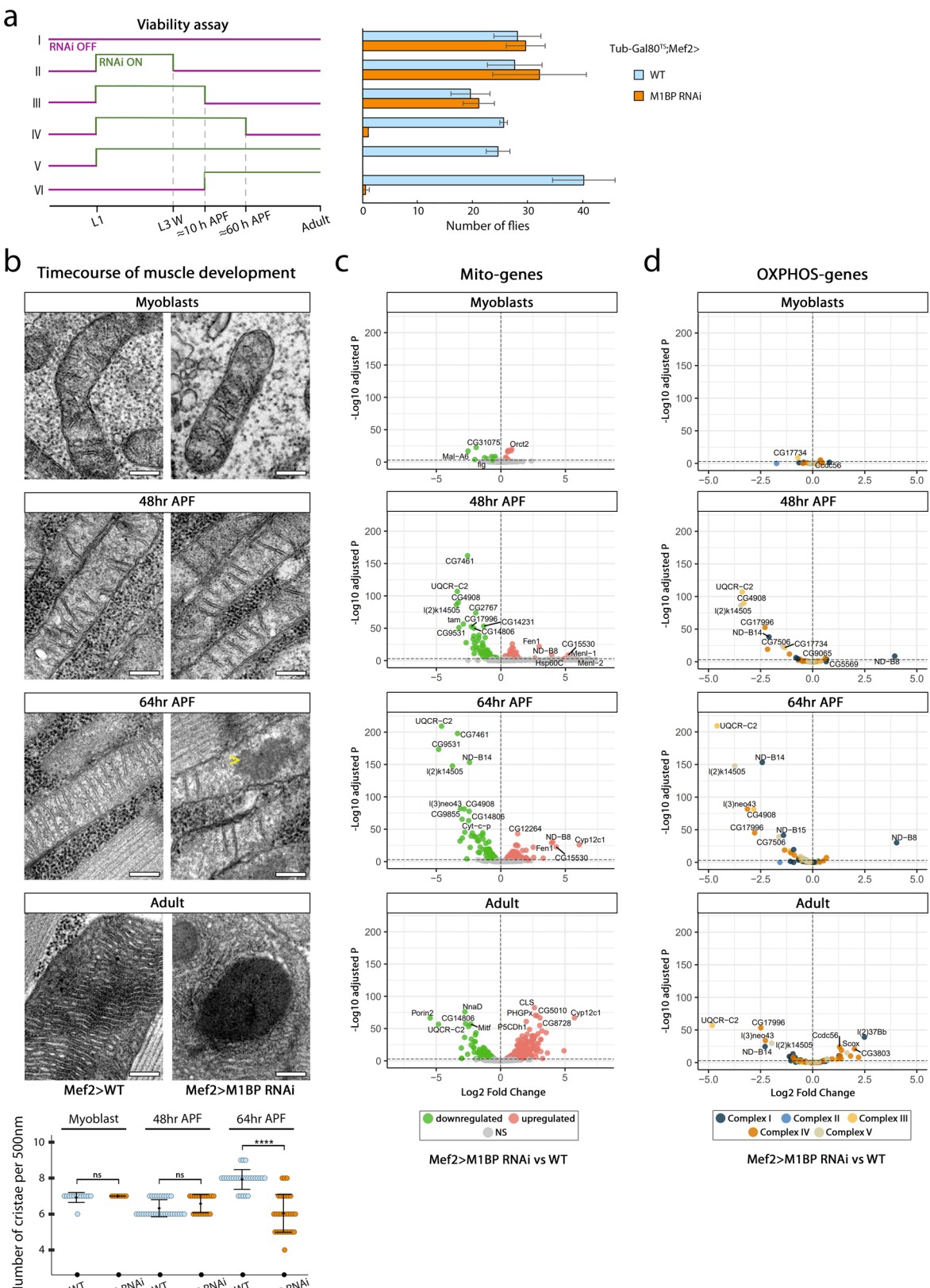

developmental time points as the electron microscopy analysis; on FACS-purified myoblasts to avoid contamination by wing disc epithelial cells and manually dissected flight muscles for pupal and adult stages. We first looked at the expression of all nuclear-encoded genes related to mitochondrial function from the curated MitoXplorer database[58] from which we excluded glycolysis-related genes and genes

encoded by the mitochondrial genome (Mito-genes; $n = 1064$) (Fig. 3c). Using a highly stringent threshold ($p$adj $\leq 0.001$), the differential gene expression (DGE) analyses showed that only 2% of Mito-genes were significantly deregulated upon *M1BP* RNAi in myoblasts, suggesting that M1BP is not involved in the transcriptional regulation of mitochondrial function in larval myoblasts (Supplementary Data 2),

**Fig. 3 | M1BP regulates the expression of OXPHOS genes during flight muscle development. a** *M1BP* downregulation was temporally controlled using *Mef2-GAL4* driven *tub-Gal80*[ts] transgene. For each condition, the mean and standard deviation of the number of adult flies (both sexes included) is represented resulting from two independent biological replicates. **b** Transmission electron microscopy micrographs of mitochondria from WT (left) and M1BP-depleted (right) myoblast and pupal flight muscles at 48 h and 64 h APF, using *Mef2-GAL4* driver. Arrowhead highlights amorphous electron-dense inclusions visible in the matrix in 64 h APF preparations upon *M1BP* RNAi. Scale bars represent 200 nm. Quantification of cristae number per 500 nm is shown for each time point ($n$ (Myo$_{WT}$) = 12, $n$ (Myo$_{RNAi}$) = 8, $n$ (48 h$_{WT}$) = 29, $n$ (48 h$_{RNAi}$) = 15, $n$ (64 h$_{WT}$) = 15, $n$ (64 h$_{RNAi}$) = 28 mitochondria). After applying Shapiro-Wilk test for normality, two-sided Mann–Whitney test was applied ($p$ values: Myo = 0.53; 48 h = 0.08; 64 h = $1.36 \times 10^{-8}$ ****). **c, d** Volcano plots of all differentially expressed nuclear-encoded mitochondrial (**c**) or only OXPHOS-related (**d**) genes between Mef2>WT and Mef2 > *M1BP* RNAi at corresponding time points. For statistical analysis, the Wald test with Benjamini–Hochberg-correction was applied. Source data are provided as a Source data file.

confirming the electron microscopy data. At 48 h APF, 10% of Mito-genes become deregulated with approximate equal numbers being upregulated or downregulated (Supplementary Data 3). At 64 h APF nearly a quarter (22%) of all Mito-genes were differentially expressed upon *M1BP* RNAi, with more genes being significantly downregulated than upregulated (Supplementary Data 4). At the adult stage, we detected 41% differentially expressed Mito-genes upon *M1BP* KD (Supplementary Data 5).

At both pupal stages as well as the adult stage, the most down-regulated genes, such as *UQCR-C2* and *l(2)k14505* are related to OXPHOS, whose expression levels were decreased by up to 30-fold. This prompted us to analyse separately genes related to OXPHOS using the updated list of OXPHOS-encoding genes we curated (Supplementary Data 1 and Fig. 3d). Interestingly, we observed a time-dependent increase in the number of OXPHOS-encoding genes that were significantly downregulated upon *M1BP* muscle-specific RNAi: whereas only two OXPHOS genes were differentially expressed at the myoblast stage, this increased to 15 genes (12%) at 48 h, 40 genes (32%) at 64 h and 35 genes (28%) at the adult stage and nearly all of these were downregulated.

The mitochondrial genome (mtDNA) encodes 13 subunits contributing to all OXPHOS complexes, except Complex II. The dynamics of their expression during myogenesis generally follows those expressed in the nucleus (Supplementary Fig. 6a, b). While intriguingly we observed an increased expression of some of the 24 tRNAs encoded by mtDNA during myogenesis, with the exception of a transient decrease in expression in *ND4L*, none of the 13 OXPHOS subunits genes encoded by the mitochondrial genome were affected by *M1BP* RNAi (Supplementary Fig. 6c and Supplementary Data 6).

Altogether, these data suggest that M1BP is required for the expression of nuclear-encoded OXPHOS genes during a critical period of their transcriptional upregulation.

### A regulatory network exists at the promoters of OXPHOS genes

Being a transcription factor, we wondered whether M1BP transcriptionally regulates the expression of other transcription factors and cofactors whose role in OXPHOS regulation in other tissues has been already established (reviewed in ref. 59), which could account for the observed downregulation of OXPHOS gene transcription following *M1BP* KD. These known OXPHOS transcriptional regulators are: Mito-chondrial transcription factor A (TFAM), a transcriptional regulator of the mitochondrial genome[60], the Rbf/E2f pathway linked with muscle metabolism and development[25,61] and known OXPHOS activators such as Erect Wing (Ewg, homologue of vertebrate NRF1)[62], Ets97D/Delg (homologue of GABPA)[63], Cap-n-collar (Cnc, homologue of NFE2L1/NRF2)[64], Spargel (Srl, homologue of PGC1-α)[65], its target Oestrogen-related receptor (ERR, homologue of ERRα/Essrg), the conserved Myocyte enhancer factor 2 (Mef2, homologue of MEF2)[66], Cyclic-AMP response element-binding protein B (CrebB, CREB1)[67], Host cell factor (Hcf), Pleiohomeotic (Pho, homologue of YY1)[68] and Stimulatory Protein 1 (SP1). With the exception of *ets97D*, *pho*, and *ERR*, available developmental transcriptome data[7] show that all factors are expressed at the mid-pupa stage of flight muscle development at levels at least equal to or higher than *M1BP* (Supplementary Fig. 7a), suggesting they could all represent potential M1BP targets in this tissue. However, upon *M1BP* RNAi, we observed no significant change in the expression of any of these genes in myoblasts and pupal stages, except for a transient decrease of *srl* levels. At the adult stage, we detected a slight downregulation of *crebB*, *mef2*, *hcf* and *ewg* levels (Supplementary Fig. 7b). This suggests that M1BP does not regulate the expression of other transcription factors, but rather cooperates with these factors to induce the large transcriptional upregulation of OXPHOS-genes during flight myogenesis.

To ascertain the possible regulatory networks underlying OXPHOS gene regulation, we undertook an in silico analysis of DNA-binding motif occurrence in the promoters of OXPHOS-genes regulated by M1BP and other OXPHOS sequence-specific transcription factors for whose DNA binding motifs were known. This approach showed that amongst the ten sequence-specific transcription factors studied, the DNA binding motif of M1BP is the most significantly represented in the promoters of OXPHOS-genes, followed by the E2F1 motif (Fig. 4a and Supplementary Data 7). Furthermore, the motif of M1BP can be found at 79 OXPHOS gene promoters (63%) placing it, along with Mef2, as one of the most overrepresented motifs at OXPHOS-gene promoters (Supplementary Data 7). Studying DNA motif co-occurrences among these ten transcription factors demonstrate a complex, but potentially interesting transcription factor regulatory network underlying the expression of OXPHOS genes (Fig. 4b and Supplementary Data 7).

Altogether, these data show that the main M1BP function in developing flight muscle mitochondria is in the transcriptional activation of OXPHOS-protein coding genes and that M1BP likely acts through the global transcriptional control of OXPHOS genes in cooperation with other known OXPHOS regulators.

### M1BP loss results in reduced OXPHOS complex assembly

We showed that *M1BP* KD in flight muscles leads on the one hand to a decrease in expression levels of OXPHOS genes and on the other, to ultrastructural abnormalities such as matrix inclusions and fewer mitochondrial cristae. Muscle mitochondrial cristae abnormalities and the presence of inclusions have both been linked with respiratory chain impairment[57]. Analysis of OXPHOS-encoding nuclear genes downregulated upon *M1BP* KD identifies both structural subunits and assembly factors of respiratory complexes, the latter being much more affected by *M1BP* KD (Fig. 5a and Supplementary Data 3–5). Indeed, the DGE analysis showed that at 48 h APF 6% of genes encoding structural mitochondrial subunits and 15% of genes encoding mitochondrial assembly factors are significantly downregulated upon *M1BP* KD (Fig. 5a, left panels). At 64 h, 27% of structural subunits and 35% of assembly factors are significantly downregulated (Fig. 5a, middle panels). At the adult stage, 11% of structural subunits are downregulated, and some 50% of assembly factors are deregulated, with 13% of genes that are downregulated and 37% upregulated (Fig. 5a, right panels). Interestingly, the structural subunits whose expression is the most significantly affected in all time points belong to Complex I except for UQCR-C2, belonging to Complex III. However, the affected assembly factors belong mainly to Complexes III, IV and V. We did not detect any major changes in the expression of OXPHOS genes (subunits or assembly factors) relating to Complex II. Together, these data demonstrate that M1BP is a major transcriptional regulator of genes

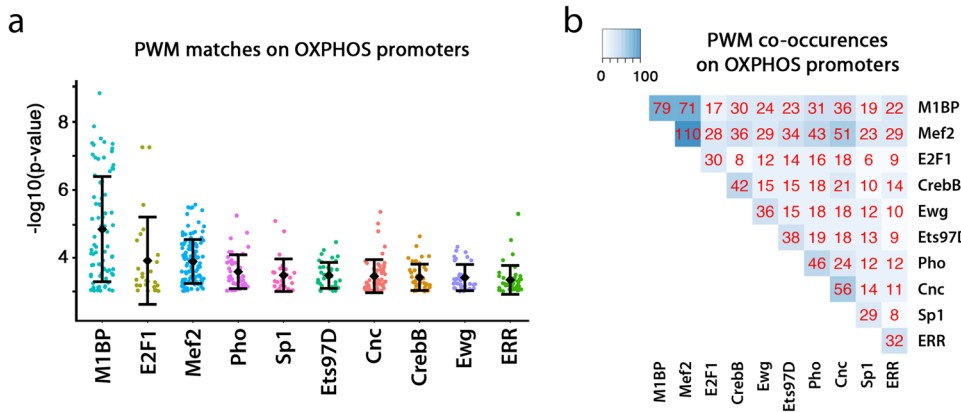

**Fig. 4 | The promoters of OXPHOS genes contain DNA binding motifs of multiple transcription factors. a** Motif occurrence analysis in promoters of OXPHOS genes of M1BP and other known OXPHOS transcriptional regulators. Each dot represents a binding motif found for a given factor in OXPHOS gene promoters (*n* (M1BP) = 80, *n* (E2F1) = 31, *n* (Mef2) = 111, *n* (Pho) = 47, *n* (Sp1) = 30, *n* (Ets97D) = 39, *n* (Cnc) = 57, *n* (CrebB) = 43, *n* (Ewg) = 37 and *n* (ERR) = 44 found motifs). The mean and standard deviation of −log10 (p-value) for each OXPHOS gene bound by a given factor is represented. *p* values, obtained using FIMO[113], are defined as the probability of a random sequence of the same length as the motif matching that position of the sequence with a score at least as good. **b** Motif co-occurrence analysis in promoters of OXPHOS genes of M1BP and other known OXPHOS transcriptional regulators, represented as a matrix, where each value corresponds to the number of genes bound by a single factor or in combination.

encoding respiratory chain assembly factors of Complexes I, III, IV, and V and structural subunits of Complexes I and III during flight muscle development.

The assembly and insertion in the inner mitochondrial membrane of OXPHOS complexes is a complex process, requiring the presence of assembly factors and a correct stoichiometry of structural subunits[69]. Having shown that M1BP KD affects the expression of numerous assembly factors and some structural subunits, we reasoned that this should lead to a decrease in the quantity of assembled OXPHOS protein complexes in the inner mitochondrial membrane (IMM). To test this, we performed Blue Native Polyacrylamide Gel Electrophoresis (BN-PAGE) on mitochondrial membrane-isolated complexes[70] which revealed that, with the exception of Complex II, for the same mitochondrial protein quantity loaded, the quantity of all mitochondrial complexes assembled in the IMM substantially decreases upon M1BP KD (Fig. 5b). Testing the in-gel enzymatic activities of Complexes I, II, IV and V (Complex III activity is incompatible with BN-PAGE[71]) demonstrated that while the assembly of Complex I, IV, and V was reduced, the assembled complexes were enzymatically active, where the difference in activity observed relative to wild type complexes is largely a reflection in the relative abundance of each assembled complex (Supplementary Fig. 8a). Interestingly, while the abundance of Complex II remained unchanged upon M1BP loss of function, the succinate dehydrogenase activity of this complex was consistently slightly higher, although the significance of this is unclear. To rule out the possibility of unequal protein loading, which could account for the observed differences, we performed the histochemical assays in a sequential manner, confirming the decreased activities of Complexes I, IV and V, while demonstrating that the increase in Complex II activity upon M1BP KD is consistent (Supplementary Fig. 8b). Altogether, these data show that the transcriptional downregulation of numerous OXPHOS assembly factors and/or structural subunits due to *M1BP* KD leads to largely reduced fully-assembled OXPHOS complexes in the adult.

Knowing that the dysfunction of the mitochondrial respiratory chain was linked with the appearance of mitochondrial inclusions[56,57], and that the subunits of OXPHOS complexes, destined to be inserted in the membrane are highly hydrophobic in nature, we hypothesised that the observed mitochondrial inclusions could contain misfolded, aggregated proteins, including OXPHOS complex components. To study this, we performed immunoblotting analyses of detergent soluble and insoluble protein fractions obtained from adult flight muscle-purified mitochondria. In line with the BN-PAGE data, using an antibody against a Complex V subunit, ATP5A we observed a decrease of ≈30% of ATP5A protein quantity in the soluble fraction upon M1BP KD. Interestingly, we observed a small amount of ATP5A in the detergent-insoluble fraction, solely in the M1BP KD condition (Fig. 5c). These data indicate that OXPHOS subunit solubility changes upon M1BP KD. Since the presence of an OXPHOS subunit in the insoluble-fraction does not demonstrate its presence in inclusions, we performed immuno-electron microscopy (Immuno-EM), allowing us to access mitochondrial inclusions on tissue sections. Quantification of ATP5A density in the cell cytosol, mitochondrial matrix or in mitochondrial electron-dense inclusions showed a significantly higher density of this subunit in mitochondrial inclusions compared to the mitochondrial matrix and outside of mitochondria (Fig. 5d). These results strongly suggest that the electron-dense inclusions represent protein aggregates, likely resulting from unassembled OXPHOS complex components.

### M1BP KD triggers a mitochondrial protein quality control response

Since most proteins operating in mitochondria are imported from the cytosol, mitochondrial function depends on the correct folding and turnover of these proteins assured by mitochondrial protein quality control (reviewed in ref. 72). In line with our hypothesis that mitochondrial inclusions represent protein aggregates, amongst the most significantly upregulated genes upon *M1BP* KD in the adult flight muscles were genes linked to a proteotoxic response, such as ubiquitin-dependent catabolic process and cellular response to unfolded proteins (Supplementary Data 8). We thus performed a differential gene expression analysis upon *M1BP* KD using GO-based lists of genes related to the unfolded protein response (UPR, 93 genes), the ubiquitin-proteasome system comprising the catalytic subunit 20 S and the regulatory particle 19S (UPS, 35 genes), and mitochondrial proteases (Mitoproteases, 10 genes) (Supplementary Data 8). We detected significant upregulation of half of all UPR-related genes, all proteasome-related genes as well as all Mitoprotease-encoding genes (Fig. 6a, lower panels). Interestingly, we were able to detect changes in the expression of these genes already at the mid-pupa stage, when mitochondrial inclusions begin to appear (Fig. 6a, upper panels). These data demonstrate that M1BP KD elicits a strong transcriptional activation of protein quality control pathways.

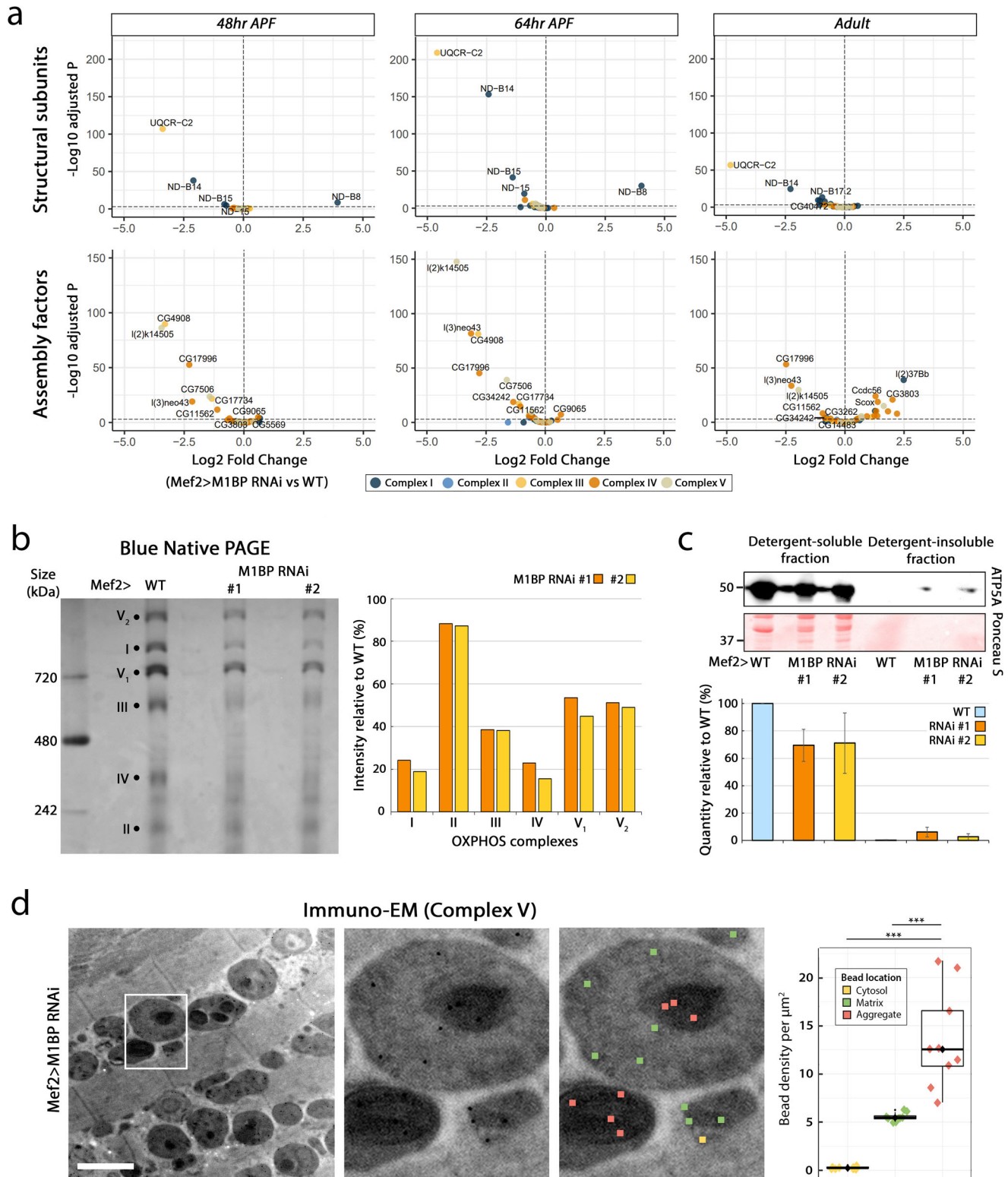

UPR is a cellular response to stress induced by the abnormal accumulation of unfolded or misfolded proteins either at the endoplasmic reticulum (UPR^ER) or in mitochondria (UPR^mt) (reviewed in[73,74]). While a large number of heat shock proteins, including Hsp70 family members, can localise to both the cytoplasm and mitochondria depending on the source of stress, we noticed among the most significantly upregulated genes the exclusively mitochondrial-localised small heat shock protein *Hsp22*[75] (Supplementary Data 6). This chaperonin interacts with mitochondrial-localised Hsp70 and Hsp60

family members[76], suggesting that the UPR observed upon *M1BP* KD is primarily of mitochondrial stress origin, which we sought to investigate in the flight muscle. Using mit::mKate2, a near far-red reporter line, to label mitochondria in fixed tissue, the Hsp70 stress response was shown to localise mainly to mitochondria (Fig. 6b), which together with the more than three-hundred-fold transcriptional increase in mitochondria-specific *Hsp22* demonstrates that *M1BP* KD leads primarily to a UPR^mt. Of note here is that like for mit::Dendra2 staining (Fig. 1c), large circular areas of the mitochondrial matrix were devoid

**Fig. 5 | M1BP KD leads to reduced mitochondrial respiratory complex assembly. a** Volcano plots of differential gene expression analyses between Mef2>WT and Mef2 > *M1BP* RNAi developing DLM. The top volcano plots represent genes encoding structural subunits and bottom plots assembly factors of mitochondria respiratory complexes. Genes are colour-coded accordingly to their complex membership. For statistical analysis, the Wald test with Benjamini–Hochberg-correction was applied. **b** Blue native polyacrylamide gel of mitochondrial complexes extracted from adult flight muscle mitochondria. Complexes are depicted by a black circle, $V_1$ and $V_2$ refer to the monomer and dimer of complex V, respectively. Quantifications of the intensity of bands, representative of the protein quantity for each complex are represented in the bar chart as a percentage relative to the WT. **c** Western Blot of ATP5A Complex V subunit on detergent-soluble and insoluble protein fractions from WT and M1BP-depleted adult purified mitochondria. The

Ponceau S membrane staining is shown as a loading control. The mean band intensities are represented as histograms for each condition from two independent biological replicates. Protein size is indicated in kDa. **d** Representative transmission electron micrographs (with an enlarged view on the right, depicted by a white rectangle) of the ImmunoEM experiment using gold-conjugated secondary antibody targeting an anti-Complex V antibody. The gold conjugates were quantified with respect to their location. Data are represented in boxplots (median and quartiles) with whiskers (minimum to maximum), where each image gave rise to a single value of the density of beads for each of the three locations; $n = 9$ sections; total number of beads= 1048 in the matrix, 169 in the cytosol and 102 in aggregates. The scale bar corresponds to 2 μm. After applying non-parametric Kruskal–Wallis test the pairwise, two-sided Mann–Whitney test was used ($p$ values = 0.0004 ***). Source data are provided as a Source data file.

of mit::mKate2 staining upon M1BP RNAi and these excluded areas did not contain Hsp70 protein (see Fig. 6b, enlarged area). It is thus possible that these areas, representing the electron-dense protein aggregates are likely inaccessible to the UPR$^{mt}$, reinforcing the notion that the observed aggregates are totally isolated from the remaining matrix by the IMM. These data showing a strong UPR$^{mt}$ were confirmed by immunoblotting analyses on purified proteins from flight muscle mitochondria, using an anti-Hsp70 antibody (Fig. 6c). Interestingly, we observed upon M1BP KD several instances of ubiquitinylated proteins on mitochondria that are very lowly or not present in the WT condition (Fig. 6c, asterisk) as well as several proteins that were already poly-ubiquitinylated in the WT condition, but whose quantity strongly increase upon M1BP KD (Fig. 6c, dagger).

The UPR and UPS are the main transduction pathways that maintain protein homoeostasis under conditions of protein misfolding and aggregation and can result in the accumulation of poly-ubiquitinated proteins which are then either targeted for degradation by the proteosome or sequestered into insoluble protein aggregates (reviewed in ref. 77). To this end, analyses of the detergent soluble and insoluble protein fractions have proven useful in studying proteostasis. Since *M1BP* KD in flight muscles elicits such a large transcriptional proteotoxic response, we thus wished to analyse the general protein status in this condition. While we detected only minor changes in the detergent-soluble protein fraction between WT and M1BP-depleted adult flight muscles, in the detergent-insoluble protein fraction we detected several proteins appearing upon *M1BP* KD that are soluble in WT (Fig. 6d, asterisks) and several proteins whose insolubility increases upon *M1BP* KD (Fig. 6d, dagger). Similar to what we observed in the mitochondrial fraction, the insoluble protein fraction also showed increased ubiquitinated proteins (Fig. 6e).

Taken together, these data demonstrate that loss of M1BP function leads to widespread mitochondrial proteotoxic stress, resulting in the accumulation of insoluble protein aggregates in the mitochondrial matrix.

## Discussion
### M1BP transcriptionally regulates OXPHOS complex assembly
Relying on aerobic glycolysis for ATP generation, myoblasts, like all stem cells generally exhibit a fragmented mitochondrial network with underdeveloped cristae, yet maintaining a baseline of OXPHOS transcription and activity[19,78,79]. At this stage, we find that M1BP function is dispensable, as we see no modification to the transcription of OXPHOS-related genes through *M1BP* KD. However, at the onset of myofibrillogenesis, when OXPHOS transcription is normally upregulated together with mitochondrial cristae biogenesis, loss of M1BP function results in perturbed cristae biogenesis and loss of the transcriptional upregulation of numerous OXPHOS complex subunits and factors required for their assembly, with transcriptional defects affecting at least one major subunit or assembly factor of Complexes I, III, IV and V. Through biochemical analyses, we showed that these transcriptional defects resulted in vastly reduced assembly

of these complexes in the IMM in adults (Fig. 5). Of note is that we observed neither transcriptional defects in factors related to Complex II nor in modification to the levels of assembled Complex II upon *M1BP* KD.

Concomitant with the loss of transcriptional upregulation of numerous OXPHOS complex subunits and assembly factors, the loss of M1BP function triggers an accumulation of electron-dense inclusions in the mitochondrial matrix, originating from the mid-pupa stage and persisting until the end of muscle differentiation (see Figs. 1d and 3b) which contain at least a subunit of the respiratory Complex V (see Fig. 5d). Knowing that the expression of numerous assembly factors is downregulated at least ten-fold upon *M1BP* RNAi, we believe this leads to a reduction in mitochondrial respiratory complexes assembly and the insoluble aggregation of their precursors. Numerous observations are in favour of this: first, subunits of the mitochondrial complexes are destined for integration in the inner mitochondrial membrane and thus contain transmembrane domains, which have hydrophobic properties rendering the proteins prone to misfolding damage. Second, many mitochondrial proteins are metastable, meaning that they aggregate when their critical concentration is achieved[80,81]. Third, in the transcriptomic time-course of flight muscle development, we observed that the transcription of OXPHOS complex assembly factors precedes the transcription of structural subunits (see Fig. 2b, c). The early presence of assembly factors in mitochondria thus seems critical to prevent misfolding and aggregation of subunits upon their arrival in mitochondria. Fourth, we observed a strong mitochondrial protein quality control response upon *M1BP* RNAi starting at the mid-pupa stage, correlating with the appearance of protein aggregates in mitochondria, as well as an increase in the amount of poly-ubiquitinylated proteins in the adult flight muscles (see Fig. 6). Fifth, it had been shown in yeast, that the failure to assemble Complex V can lead to the aggregation of structural subunits in inclusions bodies and a decrease in mitochondrial cristae quantity[57]. Based on these data we suggest that the aggregates present in the mitochondrial matrix upon M1BP downregulation contain misfolded OXPHOS complex subunits.

Apart from factors assuring protein complex assembly, the correct subunit stoichiometry is crucial for the proper complex assembly and function[82]. In mitochondria, a mismatch in the stoichiometry of nuclear-encoded and mitochondria-encoded OXPHOS subunits can lead to OXPHOS components accumulation[83]. *M1BP* KD in flight muscles leads to a decrease in expression levels of a few structural OXPHOS subunits of Complex I and III (see Fig. 5a). Namely, the UQCR-C2 subunit of Complex III was systematically the most significantly downregulated gene in our RNA-seq analyses (nearly 30-fold down-regulated). Based on this evidence we believe that the perturbed OXPHOS subunit stoichiometry, together with decreased assembly factor expression levels, account for the phenotypes we observed in M1BP-depleted muscles. All these data point towards an important role for M1BP in oxidative metabolism control through the control of the transcriptional upregulation of genes required for respiratory complex assembly and integrity.

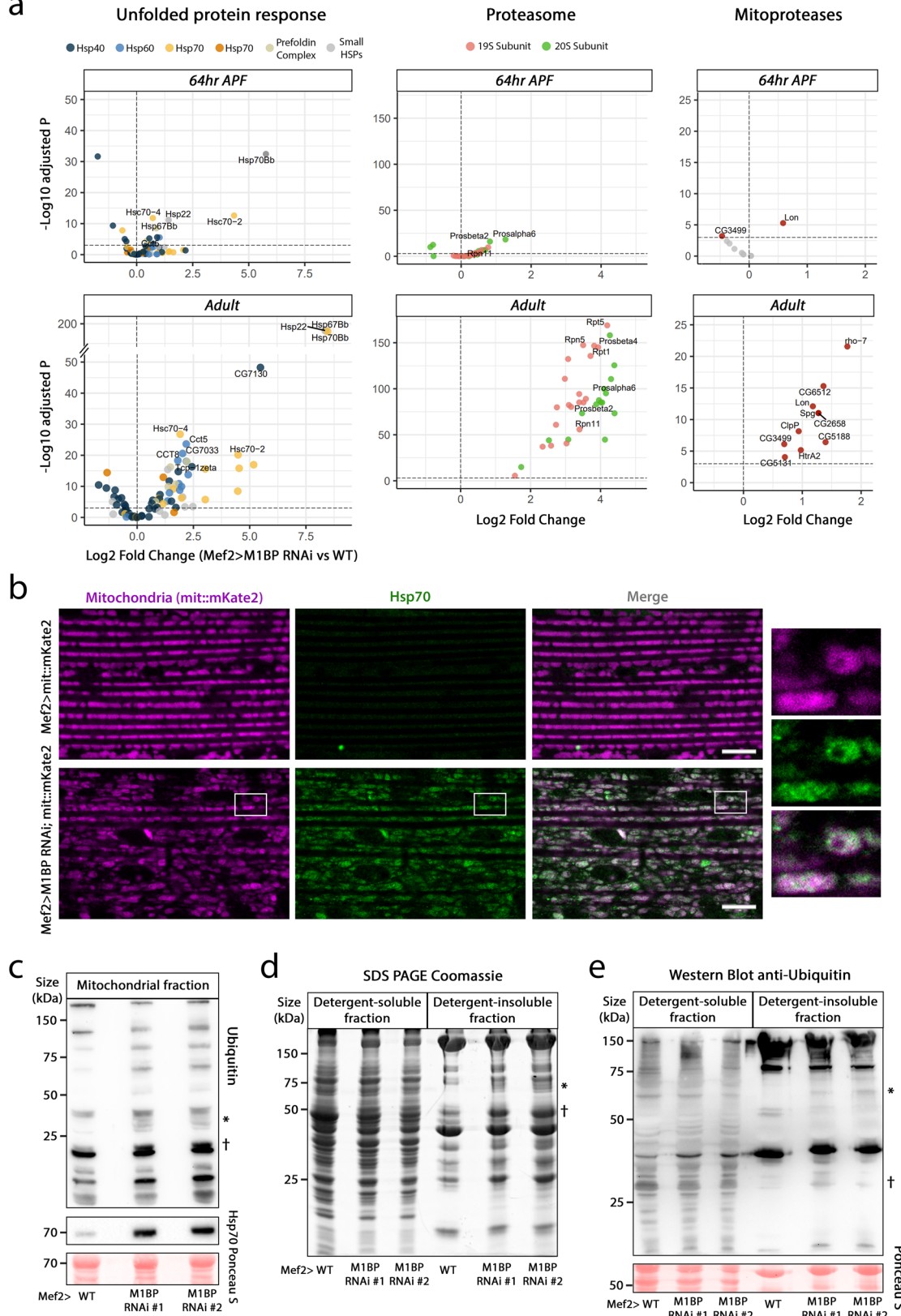

**b**

Mitochondria (mit::mKate2)      Hsp70      Merge

Mef2>mit::mKate2

Mef2>M1BP RNAi; mit::mKate2

**c** Mitochondrial fraction

**d** SDS PAGE Coomassie

**e** Western Blot anti-Ubiquitin

## M1BP and the interplay with other OXPHOS regulators

M1BP likely plays a direct role in the transcriptional upregulation of OXPHOS-related gene expression at the middle of pupation, since none of the known OXPHOS transcriptional regulators are affected by M1BP loss of function (see Supplementary Fig. 7). Noteworthy here is the E2F/Rbf pathway, where in vertebrates E2F1 is a repressor of oxidative metabolism[84] and Rbf is an activator[85]. In *Drosophila*, while E2F is an activator of the myogenic programme in flight muscle development[61], its role in oxidative metabolism is less clear. Interestingly, we showed here that the motifs of M1BP and E2F1 are the most significantly enriched motifs in the promoters of the OXPHOS-genes (see Fig. 4a). Nonetheless, Rbf, a transcriptional co-regulator that does

**Fig. 6 | M1BP KD in flight muscles triggers a widespread mitochondrial protein quality control response. a** Volcano plots of differential gene expression analysis between Mef2>WT and Mef>*M1BP* RNAi adult flight muscles. Analysed genes belong to the mitochondrial quality control response: UPR-related genes, UPS-related genes, and mitochondrial proteases. For statistical analysis, the Wald test with Benjamini–Hochberg correction was applied. **b** Confocal sections of wild-type (top) and M1BP RNAi (bottom) mitochondria in adult flight muscles labelled with mit::mKate2 and anti-Hsp70 antibody, to visualise a part of the unfolded protein response. White rectangles depict an enlarged view of a single, aggregate-containing mitochondrion (right). The scale bar corresponds to 10 μm. **c** Western blot analysis of Ubiquitin and Hsp70 on proteins obtained from adult purified mitochondria. The Ponceau S membrane staining is shown as a loading control. The asterisk represents an example of proteins with very low or no ubiquitination in the WT condition compared to the M1BP KD and the dagger a protein whose ubiquitination further increased upon M1BP KD. **d** Coomassie Blue staining of detergent-soluble and insoluble fractions obtained from adult flight muscles. **e** Western Blot analysis of Ubiquitin on detergent-soluble and insoluble fractions obtained from adult flight muscles. The Ponceau S membrane staining is shown as a loading control. Source data are provided as a Source data file.

not bind DNA directly in a sequence-specific manner, is found enriched at promoters containing Motif-1, the DNA recognition sequence of M1BP, and activates OXPHOS gene expression during myogenesis[25], suggesting that Rbf may cooperate with M1BP in the activation of OXPHOS genes in this process. Furthermore, M1BP was identified in a screen as counteracting E2F activity, similar to Rbf[86]. Interestingly, the screen by Lu and colleagues[86] also identified GFZF, a known transcriptional M1BP co-activator[87] as counteracting E2F activity. Since *GFZF* RNAi in flight muscles with *Mef2-GAL4* leads to late pupal lethality[32], it will thus be of interest to now study in detail a potential M1BP/Rbf partnership in OXPHOS gene expression, which may implicate a possible transcriptional co-activator, GFZF, in this process.

Finally, the vast majority of OXPHOS genes are ubiquitously expressed from TATA-less promoters, a characteristic of metabolic genes usually not subject to large amplitudes in gene expression[3]. This has incorrectly conveyed the idea that transcriptional regulation is not key to plasticity in OXPHOS gene expression. Yet, stem cells increase OXPHOS gene expression upon differentiation[78], somatic dedifferentiation into induced pluripotent stem cells involves the transcriptional downregulation of OXPHOS expression[88] and changes to the expression of OXPHOS genes contribute to a wide and diverse range of human diseases[89], which demonstrates the importance in defining the transcriptional mechanisms underlying such plasticity. Here, we highlight the dynamic temporal transcriptional upregulation of OXPHOS genes during myogenesis and identify M1BP as playing a fundamental role in this process. Since M1BP is a major RNA Pol II pausing factor of plethora *Drosophila* genes[26] it will now be of interest to study the interplay between the chromatin landscape of nuclear-encoded OXPHOS components, the state of paused RNA Pol II at their promoters, and the role other transcription factors and/or co-activators play with M1BP in this process as potential mediators of oxidative metabolism transcriptional plasticity.

### Similarities drawn between M1BP loss and human pathologies

The main steps and key players of muscle development are well conserved between *Drosophila* and vertebrates, and since flight muscle are not required for adult viability this makes *Drosophila* a valuable model organism for numerous human myopathies[90,91]. Flight muscles are structurally and mechanistically similar to vertebrate cardiac muscles and have been used to model human cardiomyopathies[92]. Our findings in muscle mitochondria are of clinical relevance since plethora human diseases, including multiple mitochondrial myopathies originate from mutations in genes encoding OXPHOS subunits or assembly factors[43,93]. Strikingly, some patients suffering from myopathies display mitochondrial inclusions, in some cases very similar to the phenotype triggered by *M1BP* RNAi (see[54] for examples). Mitochondrial inclusions have been also reported in patients with gastric adenocarcinomas[55]. Of interest is to note, that many patients cannot be diagnosed by light microscopy but electron microscopy is required to detect such ultrastructural mitochondrial defects[54,94].

Ultrastructural defects are due to perturbations in the highly dynamic membrane morphology of mitochondria (reviewed in refs. [95,96]). While we did not observe any transcriptional changes in the components responsible for cristae dynamics upon *M1BP* KD

during pupation, we did observe a delay in cristae remodelling and development that occurs at the onset of OXPHOS transcriptional regulation during mid-pupation (see Fig. [3]b). Furthermore, by the adult stage, the rapidly developing IMM had completely isolated the large protein aggregates from the remaining matrix space (Fig. [1]e). Isolation of protein aggregates in the matrix by the IMM represents a previously undocumented mitochondrial stress response mechanism. Understanding the molecular mechanisms driving this remodelling may be of therapeutic benefit in combating diseases resulting from mitochondrial-targeted protein aggregation-related neurodegenerative diseases, such as Parkinson's, Alzheimer's and Huntington's diseases. Indeed, such diseases often elicit a strong UPR^mt, which we also observed during myogenesis in the M1BP KD muscle (see Fig. [6]). Like in these neurodegenerative diseases, we hypothesise this proteotoxic response is the consequence of the toxicity of protein aggregates present in the mitochondrial matrix, although in the case of *M1BP* KD, we favour this is due to the primary defects observed on OXPHOS complex assembly. Nonetheless, as in the case of yeast protein aggregates upon Complex V mis-assembly[57], it remains an open question as to why, despite a strong and early protein quality control response, the observed aggregates cannot be resolved and cleared.

Finally, mitochondrial proteases are not only central regulators of mitochondrial proteostasis but play a key role in the response to mitochondrial stress. We observed all mitoproteases transcriptionally upregulated upon *M1BP* KD, the first of which, Lon protease (see Fig. [6]) is known to help mediate the turnover of non-assembled proteins in mitochondria (reviewed in ref. [97]). In addition to its protease activity, Lon protease also exhibits chaperone-like activity, working with mitochondrial heat shock proteins to exhibit anti-aggregation activities[98,99]. The observed mitoprotease response, combined with heat shock protein UPR^mt, are thus likely to represent an initial anti-aggregation response, followed by a much later protein degradation response through mitoprotease activity and ubiquitin-dependent proteasome degradation (Fig. [6]). Given that numerous mitoproteases are transcriptionally upregulated in human diseases, including highly aggressive metastatic cancers in the case of Lon (reviewed in refs. [100,101]), it will be of benefit to use M1BP KD in flight muscle as a means to better study and characterise the mitoproteases in the mitochondrial stress response, particularly pertaining to human diseases.

## Methods

### Fly stocks

Flies were raised under standard conditions at 25 °C, unless otherwise stated, in a 12 h light/12 h dark cycle. For *M1BP* downregulation, the following lines were used: UAS-*M1BP* RNAi #1 (110498/KK VDRC) and UAS-*M1BP* RNAi #2 (BL32858). All experiments were performed with both RNAi lines and if unspecified, including for RNA-seq experiments, data for the weaker RNAi #1 line are shown.

GAL4 drivers used in this study were muscle-specific *Mef2-GAL4* (BL27390) and *Him-GAL4*[7], pan-neuronal *elav-GAL4* line (BL458, C155) and fat body-specific *cg-GAL4* line (BL7011). For all experiments performed on myoblasts, we staged larvae to 0 h APF (white pupa), where the number of larval myoblast is the highest[102]. Myoblasts were visualised using a *twi::GFP* transgene (BL79615). Lines allowing

mitochondria imaging in flight muscles were *UAS-mit*::Dendra2[39] and *UAS-mit*::*mKate2* (chr. 2 and chr. 3) that was generated by P-element-mediated transgenesis using *pUASp* and contains the far-red fluorescent protein mKate2 (excitation 588 nm, emission 633 nm) fused at the N-terminus to human COXVIII mitochondrial target sequence (MSVLTPLLLRGLTGSARRLPV PRAKIHSL).

For conditional *M1BP* depletion, the TARGET system was used[53]. In this system, *TubGAL80*[TS] is expressed at 18 °C, repressing *Mef2-GAL4* activity and is inactivated at 29 °C, allowing for *Mef2-GAL4* to be active, allowing *UAS-RNAi* expression. Females laid eggs for a period of 24 h at 18 °C and one hundred embryos per genotype and condition were collected. Eggs were kept at 18 °C during the whole embryogenesis and then RNAi activated through switching to 29 °C at either larval L1 or early pupa stage (≈10 h APF). RNAi was then either maintained until adult or removed by switching to 18 °C at L3 wandering, early (≈10 h APF) or mid-pupa (≈60 h APF) stages. Eggs kept at 18 °C during whole embryogenesis served as a control. The number of eclosed flies for each condition was quantified.

## Immunofluorescence

**Imaginal discs.** White pupae (0 h APF) were dissected in PBS and fixed for 20 min in 4% formaldehyde in PBS, rinsed and washed three times in PBT (0.2% Triton X-100, PBS), 10 min each. Samples were blocked in 4% bovine serum PBT for at least 1 h and incubated with primary antibodies in 4% bovine serum in PBT overnight (ON) at 4 °C. After three washes in PBT samples were incubated with secondary antibodies for 1h30 followed by washing six times in PBT for 10 min each. Imaginal wing discs were dissected using forceps (Fine Science Tools, Dumont #5), mounted in a Vectashield medium containing DAPI (H-1200, Vector Laboratories) and imaged using a confocal microscope (Zeiss LSM 880). All steps were performed at room temperature, using gentle agitation, unless otherwise stated. Primary antibodies were rabbit anti-M1BP (1/250, D. Gilmour) and chicken anti-GFP (1/1000, GFP-1010, AB_2307313, AvesLabs). Secondary antibodies were AlexaFluor 488, donkey anti-chicken A10039, Invitrogen, 1:500 and AlexaFluor 568, goat anti-mouse, Invitrogen, A11004, 1:500.

**Indirect flight muscles.** Adult females were quickly rinsed in ethanol, and wings, head and abdomen were removed in PBS. Thoraces were split in half using forceps and fixed in 4% formaldehyde in PBS for 20 min. Hemithoraces were rinsed and washed three times in PBT (0.5% Triton X-100, PBS) for 10 min each and blocked in 3% NGS (S26-M, Sigma-Aldrich) in PBT for at least 1 h. Samples were incubated ON with primary antibodies at 4 °C. Samples were rinsed and washed three times in PBT for 15 min each, incubated with secondary antibodies including Phalloidin (1/200, Phalloidin-Atto 647 N, Sigma-Aldrich) and Hoechst (1/1000) at RT for 1h30 and washed again in PBT six times for 10 min each. Hemithoraces were mounted in a Vectashield medium (Vector Laboratories) and imaged using a confocal microscope (Zeiss LSM 880). All steps were performed in a 24-well plate under agitation. Primary antibodies used were rabbit anti-M1BP (1/250, gift from D. Gilmour), and mouse anti-ATP5A (1/250, 15H4C4 Thermo Scientific 43-9800). In Fig. 6, mitochondria were visualised using *Mef2-GAL4* driven *UAS-mit*::*mKate2* line and UPR was visualised with mouse anti-HSP70 antibody (1/100, StressMarq Biosciences Cat# SMC-106, RRID: AB_2295500). Secondary antibodies were AlexaFluor 488, goat anti-rabbit, Invitrogen, A11034, 1:500 and AlexaFluor 568, goat anti-mouse, Invitrogen, A11004, 1:500.

## Live flight muscle imaging

For adult flight muscle imaging, adult females expressing *UAS-mit*::*Dendra2* under the control of the *Mef2-GAL4* driver were briefly rinsed in ethanol and transferred to dissecting plate containing PBS. The head, abdomen, wings, legs and thorax ventral cuticle were gently removed using forceps. Thorax was split with forceps in two hemithoraces that

were quickly transferred to a slide containing PBS. Three layers of double-sided tape were used to increase the slide height and prevent tissue deformation. Hemithoraces were imaged using a confocal microscope (Zeiss LSM 880).

## Transmission electron microscopy

For myoblast imaging, wing discs of 0 h APF pupa were fixed for 2 h in a fixation solution (2.5% glutaraldehyde, 2% paraformaldehyde and 0.1 M sodium cacodylate pH 7.3), embedded in 4% low melting agarose (A9414, Sigma) and kept at 4 °C in the fixation solution until further processing. For pupal flight muscle visualisation, pupa were staged at 25 °C, removed from the pupal case and poked twice in the abdomen with a microneedle for the penetration of the fixation solution. The abdomen has been removed to allow for proper osmium penetration. For adult flight muscles, adult males were quickly rinsed in ethanol and dissected in PBS. The head, wings, abdomen and legs were removed, as well as the ventral thoracic cuticle. Thoraces were fixed in a fixation solution and kept at 4 °C until further processing. Samples were further processed according to the published protocol[21] and imaged on a Transmission Electron microscope FEI Tecnai G2 at 200 keV.

For tomographic reconstruction of mitochondrial aggregates, serial 350nm-thick sections were deposited on slot grids and contrasted in 2% uranyl acetate for 20 min and lead citrate for 8 min. Dual-axis tomographic acquisitions were carried out in an FEI Tecnai G2 at 200 keV using a Veleta camera (Olympus, Japan) for image collection from −60° to +60°. Tomographic reconstructions and tomograms joining were carried out in eTomo (IMOD, version 4.11), segmentation was carried out in Ilastik (version 1.4.0) and surface rendering in Amira (version 2022.2). Observed reconstructions represent 75% certainty, due to missing upper and lower wedges that are inherent in the protocol.

For the Immuno-EM experiments, 2-day-old females were processed as for transmission electron microscopy imaging above, until the dehydration step. Sample were embedded in epoxy resin and cut in 90 nm and deposited on nickel mesh grids. Grids were treated with a saturated solution of Sodium Meta-Periodate (S-1878, Sigma) for 2 min, washed with 1% Triton X-100 in TBS for 5 min, saturated 1 h in 5% BSA, 0.5% Fish Gelatine skin and incubated ON with mouse anti-ATP5A (1/100, 15H4C4 Thermo Scientific 43-9800). Grids were washed 4 times in TBS for 5 min each, incubated with an anti-mouse secondary antibody conjugated with 10 nm gold (1/30, Aurion) in TBS at 37 °C for 1 h, washed 4 times in TBS for 5 min each and postfixed in 2.5% glutaraldehyde, 0.05 M Sodium Cacodylate for 10 min. Acquisitions were carried out in an FEI Tecnai G2 at 200 keV using a Veleta camera (Olympus, Japan).

## Muscle locomotor assays

To assess flight capacity, 20–30 3–4-day-old flies per genotype (of separated sex) were thrown into a Plexiglas cylinder (with the height of 1 m and a diameter of 8.4 cm), and the landing position in the cylinder (top, middle, bottom) was scored[103]. Flies landing at the top are considered as able to fly, in the middle as weak flyers and to the bottom as flightless. At least 90 flies were tested per genotype (except for *M1BP* RNAi #2 leading to significant adult lethality) and at least two independent biological replicates were performed. To test the flight muscle fatigability, flies were tapped 3 times and let to recover for a few seconds. This was repeated three times and directly after that, flies were subjected to the flight test. Negative geotaxis assay was performed as previously described in ref. 39. The walking speed was determined by scoring the distance travelled (in cm) in 10 s, repeated five times.

## Mitochondria purification

Mitochondria purification was performed according to a modified procedure[104]. Flight muscles from 15–20 adult female flies were

 **14**

dissected in ice-cold PBS and crushed with a Potter homogeniser in the mitochondrial isolation buffer containing 250 mM sucrose and 0.15 mM $MgCl_2$ in 10 mM Tris-HCl, pH 7.4, on ice. Muscle homogenate was centrifuged twice at 400 g for 5 min at 4 °C to remove insoluble material. The supernatant was centrifuged at $7000 \times g$ for 5 min at 4 °C. The mitochondrial pellet was washed twice with the mitochondria isolation buffer and stored at −80 °C. Mitochondrial protein concentrations were determined using the Bradford method[105].

## Blue native PAGE (BN-PAGE) analysis and in-gel activity assay

BN-PAGE was performed using NativePAGE Bis-Tris gels, following manufacturer protocol (Invitrogen). 20 μg of mitochondria was solubilised in a buffer containing native PAGE sample buffer (BN2003, Invitrogen) supplemented with 1% of digitonin (digitonin/protein (w/w) ratio of 8) (D5628, Sigma) and EDTA-free Protease inhibitors (11873580001, Sigma) and incubated on ice for 20 min. After centrifugation at $20,000 \times g$ for 30 min at 4 °C, the supernatant was recovered and mixed with the 5% G-250 Sample Additive (BN2004, Invitrogen). The final volume was loaded in 3–12% Bis-Tris NativePAGE gels (BN2012BX10, Invitrogen) and the electrophoresis was performed using the NativePAGE Running buffer (BN2001, Invitrogen) as anode buffer, and the NativePAGE Running buffer containing the NativePAGE Cathode Additive (BN2002, Invitrogen), as cathode buffer. NativeMark Protein standard (LC0725, Invitrogen) was used as the molecular weight marker, and run with the samples. Following electrophoresis, the resulting gel was either stained with Coomassie Brilliant blue G 250 (1.15444, Sigma) or was subjected to in-gel activity assays as described below[106,107]. Complexes were optionally analysed on separate gel stripes or consecutively. Source data are provided as a Source Data File.

Complex I activity was performed by incubating the native gels in a solution containing 2 mM Tris-HCl (pH 7.4), 0.1 mg/ml NADH and 2.5 mg/ml Nitrotetrazolium Blue chloride for 20 min. After the appearance of violet bands, the reaction was stopped by the addition of 10% Acetic Acid. The Complex II in-gel activity assay was performed by incubating the gel in a solution containing 5 mM Tris-HCl (pH 7.4), 20 mM Sodium Succinate, 2.5 mg/ml Nitrotetrazolium Blue chloride and 0.2 mM Phenazine Methosulfate for 40 min. After the appearance of the violet colour indicative of Complex II activity, the reaction was stopped with 10% acetic acid.

The Complex IV in-gel activity assay was performed by incubating the gel in a solution containing 0.5 mg/ml 3,3′-diaminobenzidine (DAB), 1 mg/ml of the Complex IV substrate cytochrome $c$ and 50 mM Phosphate buffer (pH 7.4) for 40 min. After the brown product appearance, the reaction was stopped with 10% acetic acid.

The Complex V in-gel activity assay was carried out by pre-equilibrating the gel in a solution containing 40 mM Tris-HCl (pH 8.5), 4 mM ATP, and 1.5 mM magnesium chloride supplemented with 30 mM of N-dodecyl-β-D-maltoside (DDM) for 3 h. After removal of the solution, the gel was washed for 1 min in water and incubated in a buffer consisting of 35 mM Tris-HCl, 270 mM glycine, 14 mM magnesium sulfate, 0.075% lead nitrate, and 0.8 mM ATP (pH 7.8) with 20% methanol for 40 to 60 min. Following the appearance of white bands, the reaction was stopped with 50% methanol. The activity of Complex III cannot be visualised on the Blue Native PAGE gel[71]. Source data for complex activities are provided In the Supplementary information file.

## Protein solubility fractionation analyses

Detergent-soluble and insoluble protein fractions were obtained from adult flight muscles or purified mitochondria (see "Mitochondria purification" section) following a published protocol[108] with minor modifications. Adult thoraces/mitochondria were homogenised using 0.5 mm silica glass beads (Beadbug, Z763748-50EA) in a Precelyss 24 tissue homogeniser (Bertin technologies) at 5500 rpm, 2 × 25 sec with 40 sec pause. Protein fractions or purified mitochondria were mixed with Laemmli buffer, denatured for 5 min at 95 °C and run on two 12% SDS-polyacrylamide gels. One of the gels was stained with Coomassie Brilliant blue G 250 (1.15444, Sigma) and the other subjected to western blotting using standard protocols. Primary antibodies used were mouse anti-ATP5A (1/1000, 15H4C4 Thermo Scientific 43–9800), rabbit anti-Ubiquitin polyclonal antibody (1/100, U-5379 Sigma) and mouse anti-HSP70 antibody (1/1000, SMC-106). Secondary antibodies used were light chain-specific, conjugated to HRP, anti-rabbit-HRP (1/20,000, 211-032-171, Jackson ImmunoResearch) and anti-mouse-HRP (1/20,000, 115-035-174, Jackon ImmunoResearch). Source data are provided as a Source data file.

## Real-time quantitative PCR

Total RNA was extracted from 10 adult females using Promega kit (ReliaPrep™ RNA Miniprep Systems, Z6111), according to the manufacturer specification, except the initial centrifugation step replaced by 1 min vortex and 5 min incubation at RT. Extracted RNA was reverse transcribed using Superscript Reverse Transcriptase IV kit (#18090010, Invitrogen). The relative mRNA level was quantified using SYBR Green qPCR Master Mix (#4472954, Invitrogen) in a CFX96 BioRad Detection system. The reference genes *pfk*, *ald*, and *gapdh* were validated as stable control genes. Two to three independent biological replicates per condition were performed. Primers used in the experiment are: for *pfk* (forward 5′-TCGATGCCATCTCCAGTACA-3′, reverse 5′-CCCAT GACCTCCATGATGA-3′), for *ald* (forward 5′-AATGACGACCTACTTC AACTACCC-3′, reverse 5′-AACGATTTTCTGGGCGATT-3′), for *gapdh* (forward 5′-AGCGAACTGAAACTGAACGAG-3′, reverse 5′-AATTCCGA TCTTCGACATGG-3′) and for *M1BP* (forward (5′-CGCATGGCCTTTGA ACTT-3′, reverse 5′-GAAGCGCGACTGACAGATT-3′).

## Sample preparation for RNA-seq

For the wing disc-associated myoblasts dissociation was performed according to a modified protocol[109]. 0 h APF *Mef2-GAL4*, *UAS-GFP::Gma*/WT or *UAS-M1BP* RNAi #1 expressing larvae were washed in PBS, sterilised with 70% ethanol and dissected in Supplemented Schneider's Medium (SSM) containing 10% foetal bovine serum, 2% Penicillin/Streptomycin and 0.02 mg/ml Insulin. 30 to 35 wing discs were collected in ice-cold PBS and dissociated using preheated TrypLE 10× at 30 °C for 45 min at 300 rpm. The reaction was stopped by adding SSM and the cells were mechanically resuspended by gentle pipetting. Aggregates were removed by filtering cells in a cell strainer (30 μm, Miltenyi) and cells were sorted using a FACS AriaII Machine (BD), according to cell viability, size and GFP intensity. Myoblasts were sorted directly in the kit-provided Lysis buffer supplemented with 2-Mercaptoethanol (#M6250, Sigma) and kept at −80 °C until RNA extraction, which was preferentially performed the same day. RNA was isolated from the pure myoblast population of approximately 50,000 cells using RNA extraction Microkit (#74004, Qiagen).

For pupal flight muscle RNA-seq, *Mef2-GAL4*/WT or *UAS-M1BP* RNAi expressing pupa were staged at 25 °C and dissected in ice-cold PBS. The dissection was performed according to a published protocol[103]. Hand-isolated flight muscles from 6 to 8 pupa were directly collected in the kit-provided Lysis buffer supplemented with 2-Mercaptoethanol (#M6250, Sigma), homogenised with a small pestle and RNA was extracted using RNA extraction Microkit (#74004, Qiagen).

For adult flight muscles, RNA isolation was performed on hand dissected indirect flight muscles from 30, 2-days old females. The tissue was collected in batches of 15 flies at a time to minimise the time between samples and TRIzol treatment and transferred to Eppendorf tubes, then centrifuged at $2000 \times g$ for 5 min at 4 °C. The flight muscle pellet was resuspended in TRIzol reagent (Thermofisher cat #15596026), shock froze in liquid nitrogen and kept at −80 °C. RNA was isolated directly from the TRIzol muscle samples using a tube-based purification kit from Zymo Research (Direct-zol RNA Microprep, Zymo Research, #R2060).

For all samples (myoblast, pupa and adult flight muscles) RNA quality was verified via Bioanalyzer (Agilent).

## High-throughput sequencing analysis

The analyses of mito-gene and sarcomeric gene expression during flight muscle development presented in Fig. 2 and Supplementary Fig. 5 were performed on DESeq2-normalised RNA-sequencing data provided from ref. 7.

RNA sequencing was performed on libraries prepared from triplicate RNA preparations. Libraries for myoblast preparations were performed using the SMART-SeqX v4 UltraX Low Input RNA Kit for Sequencing (Takara Bio Europe, Saint Germain en Laye, France) according to manufacturer's instructions with 10 cycles of PCR for cDNA amplification by Seq-Amp polymerase. Six hundreds pg of pre-amplified cDNA were then used as input for Tn5 transposon tagmentation by the Nextera XT DNA Library Preparation Kit (96 samples) (Illumina, San Diego, USA). Libraries from DLM preparations were performed using the TruSeq stranded mRNA Sample Preparation Kit and TruSeq RNA Single Indexes kits A and B (Illumina, San Diego, CA) following the manufacturer's instructions. All library preparations were followed by 12 cycles of library amplification, further purified with SPRIselect beads to remove primer adaptors and size selection and validated for concentration and fragment size using Agilent DNA1000 chips. Sequencing was performed using a HiSeq 4000. Base calling performed using RTA (Illumina) and quality control performed using FastQC.

Differentially expressed genes were called using DESeq2[110] (version 1.34.0) using a false discovery rate (p-adjusted value in DESeq2) threshold of 0.001 and having a 1.5× fold change in expression upon *M1BP* RNAi. The lists of differentially expressed genes for all RNA-seq datasets are available in Supplementary Data 2–5. Gene ontology analyses were performed using FlyEnrichr[111] on differentially expressed genes, retaining the top ten ontologies having an adjusted *p* value of less than 0.01.

## Identification of transcription factor binding sites on OXPHOS promoters

Position weight matrices (PWMs) of transcription factor binding sites were obtained from JASPAR[112]. For *Drosophila* transcription factors that have no identified PWM, we used the PWMs from the human or mouse homologues. The following position weight matrices were used: Drosophila M1BP (MA1459.1), human MEF2A (MA0052.4), human E2F1 (MA0024.3), human CREB1 (MA0018.4), human NRF1 (MA0506.1), human GABPA (MA0062.3), Drosophila Pho (MA1460.1, Drosophila cnc (MA0530.1), Human SP1 (MA0079.3), Mouse Essrg (MA0643.1). The obtained PWMs were used to scan OXPHOS gene promoters using FIMO (version 5.5.1)[113] using a p-value threshold of 0.001, and retaining the most significant occurrence in case of multiple matches. OXPHOS promoters (dm6 genome release) were defined as sequences encompassing 250 bp upstream and 50 bp downstream from the transcription start site.

## 3D nuclei counting

Automatic 3D nuclei counting from fluorescent images was achieved using the Python package Stardist[114,115], which implements a deep learning approach based on star-convex shape representations for object detection and comes along with different pre-trained 2D and 3D models. As the counting using the pre-trained Stardist 3D model was not accurate enough for our images, we trained a custom Stardist 3D model from scratch. We built an in-house dataset consisting of both a single volume of size 97 × 256 × 256 extracted from one of our fluorescent images and its corresponding manual annotation in 3D we created using Napari's labelling tool[116]. In order to increase the number of images, we tiled this volume into four non-overlapping sub-volumes of size 97 × 128 × 128 and used three of them for training the model and the remaining one for evaluation using the Intersection Over Union metrics.

The hyper-parameters of Stardist were left as default and 3D UNET architecture used as backend. Image normalisation (between 1 and 99.8) was applied by channel independently. Empirical object anisotropy estimations was calculated using Stardist function calculate_extents and the subsampling grid size (2,1,1) for prediction was inferenced from this anisotropy value. Ray value ($v = 96$) was left as default, train patch size (48x96x96) and train batch size ($n = 32$) was chosen accordingly with our dataset size and hardware memory constraints. Moreover, we used the data augmentation strategy proposed by Stardist consisting of random image flip, rotation and random image intensity variation. We trained the model for 80 epochs using an Nvidia GV100 GPU and monitored the training until it reached a plateau (IoU: 0.53 on validation set). Then, the optimised probability threshold was automatically inferenced by Stardist using different Non-Maximum-Suppression threshold (NMS 0.3, 0.4 and 0.5). A memory efficient inference pipeline was created to handle the prediction of 27 full stack images of size 2048 × 2048 and varying depth. All codes (training/inference) and trained models are available on github (https://doi.org/10.5281/zenodo.7695893)[117].

## Statistics and reproducibility

All experiments were performed in three independent biological replicates unless otherwise stated in the figure legend. All image processing and quantifications were performed using Fiji software[118]. The intensity of M1BP staining in myoblasts was quantified as a ratio of the M1BP intensity in the myoblast area (labelled by *twi::GFP*) and in the disc epithelium, measured using ImageJ. In adult flight muscles, the intensity of M1BP staining was measured using ImageJ with ROI applied on nuclei. Muscle parameters, sarcomere length and myofibril width were estimated using the MyofibrilJ plugin for ImageJ (https://imagej.net/MyofibrilJ)[7]. For mitochondrial quantification from fluorescence images, the Watershed process was applied on by-default thresholded images. Analyze particles tool was applied to acquire mitochondrial area and aspect ratio (defined as ratio of the major to the minor axis). Mitochondrial density and content were quantified by dividing the number of mitochondria and total mitochondrial area, respectively, by the confocal section area. For mitochondria quantification from TEM images, mitochondria were manually outlined in ImageJ and Measure tool was applied to obtain Area, Aspect Ratio, Perimeter and Circularity (defined as $\frac{4\pi \times \text{Area}}{\text{Perimeter}^2}$) values. The number of cristae was quantified manually using ImageJ over a distance of 500 nm. All BN-PAGE gels and SDS-PAGE gels were quantified from scanned gels using ImageJ. The Immunogold experiment was quantified using the Cell counter plugin (https://imagej.net/plugins/cell-counter) as a density (the number of beads per μm²) of mitochondrial matrix area, aggregate area or area outside of mitochondria. All statistical analyses applied can be found in figure legends. Statistical tests and charts were performed using R (version 4.1.0)[119,120].

## Reporting summary

Further information on research design is available in the Nature Portfolio Reporting Summary linked to this article.

# Data availability

All high-throughput sequencing data generated in this study have been deposited with the Gene Expression Omnibus (GEO) and are available under the accession number GSE207241. Source data are provided with this paper.

# Code availability

All custom code for nuclei counting of larval myoblasts in 3D space described in this manuscript are made available as open source code under GNU General Public Licence v3.0 and are available on GitHUB and accessible at https://doi.org/10.5281/zenodo.7695893.

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

## Acknowledgements

We thank Bloomington and Vienna Stock Centres for fly stocks, D. Gilmour for M1BP antibodies and the IBDM Imaging and *Drosophila* platforms. We acknowledge the PICsL-FBI Marseille imaging facility, member of the national infrastructure France-BioImaging supported by the French National Research Agency (ANR-10-INBS-04). We thank the GenomEast (IGBMC, France) sequencing platform for generation of the high-throughput RNA sequencing data. This work was supported by the Fondation ARC pour la recherche sur le cancer (PJA20181208014, A.J.S.), la Ligue Nationale Contre le Cancer (M.B.) and AFM-Téléthon (no. 23721, A.J.S.).

## Author contributions

G.P., M.B and A.J.S. conceived the project, G.P. and M.B. performed experiments and analysed data with support from T.R., who performed adult live mitochondrial imaging and Immuno-EM imaging and quantification. N.M.L. performed adult flight muscle dissection for RNA-seq. A.A. prepared all transmission electron-microscopy samples. N.B. performed tomography imaging and 3D reconstruction. F.R. performed the

Immuno-EM experiment. F.D. performed the machine-learning-based strategy for myoblast counting. C.M.-Z. performed cristae quantification for adult mitochondria. F.S. and Y.G. analysed data and provided expert experimental suggestions. A.J.S. and G.P. carried out the bioinformatics analysis of RNA-seq datasets. G.P. and A.J.S. wrote the manuscript with inputs from all authors.

## Competing interests

The authors declare no competing interests.
