## [Peer Review File · Nature Communications]

M1BP is an essential transcriptional activator of oxidative metabolism during *Drosophila* developmentREVIEWER COMMENTS

Reviewer #1 (Remarks to the Author):

The authors investigate the role of M1BP in development of flight muscle mitochondrial structure and composition demonstrating that loss of M1BP leads to reduced cristae, OxPhos complexes, abnormally shaped mitochondria, up regulation of the protein quality control response, and a flightless phenotype. Developmental regulation of mitochondrial ultrastructure in muscles is an area of great interest. However, there are concerns with the generalizability of these flight muscle data as well as several areas that should be better clarified.

It is unclear how applicable these results are beyond *Drosophila* IFMs. IFMs are asynchronous, have very little sarcoplasmic reticulum, and have individual myofibrils which run the length of the cell, all of which are unlike any known mammalian muscles or the majority of other muscles within the fly. Does the role of M1BP extend to any other muscle types? Assessment of some type of tubular muscle (e.g. leg, TDT) should minimally be performed in order to generalize these results. Additionally, discussion of the mammalian ortholog(s) of M1BP and their known role in muscle development should be addressed.

Does M1BP overexpression result in the opposite effect with increased OxPhos complexes and cristae?

Do adenine nucleotide translocase (ANT) and the Pi transporter, both key OxPhos components, follow a similar expression pattern as Complexes I-V?

Intro, third paragraph: It is well known that both fast and slow-twitch muscles can be oxidative and can have similar mitochondrial content, individual mitochondrial morphology, and mitochondrial network morphology. Discussion of muscle fiber type should discriminate between metabolic and contractile fiber types. Moreover, the short disconnected appearance of glycolytic muscle mitochondria is largely due to 2D imaging in the longitudinal axis. 3D imaging or 2D imaging along the lateral axis shows that glycolytic mitochondria in mammals are also elongated and connected along the lateral axis (e.g. Ogata and Yamasaki 1985, Picard et al 2013, Bleck et al. 2018).

Figure 1C: As stated in the manuscript, development of mitochondrial ultrastructure is critical to mitochondrial function. However, only cristae number is quantified despite the high quality of the majority of the available images. Mitochondrial shape is alluded to several times but should be backed by quantitative assessment. Mitochondrial shape (circularity, aspect ratio, etc), area, perimeter, surface area to volume, length, and cristae width are all informative measures of mitochondrial structural capacity that can be assessed in the images as shown. Mitochondrial content would also be informative and can be easily assessed in the images in Supplemental Figure 1.

Lines 162-6: It is shown that mitochondrial cristae biogenesis continue through eclosion and is speculated that biogenesis stops "the moment muscles are solicited for flight." This should be tested by assessing muscles at a second adult time point.

Figure 1 and Figure 2A,C-E use myoblast/APF/adult time points whereas Figure 2B uses embryo, larva, pupa, adult. Timing descriptors should be consistent across figures to ease comparison of data.

Figure 2B: Does the lethal M1BP phenotype require the absence of M1BP during the larval stage? It is suggested that the early/mid pupal stage is when M1BP is essential. However, in the current data in Figure 2B M1BP, is always knocked down during the larval stage first. M1BP knockdown should be restricted to the early/mid pupal stage and the phenotype assessed to strengthen the claims made. Later experiments show that knockdown from the mid-pupal stage on (Act88F-Gal4) have no phenotype, so this needs to be clarified.

Figure 2C: It is unclear at what timepoint(s) M1BP was knocked down in these experiments.

Figure 3A: The M1BP driven change in mitochondrial structure independent of changes in sarcomere structure seem to argue against the mechanical feedback model where mitochondria and myofibril biogenesis are coordinated together recently proposed by some of these authors (Avellaneda et al, 2021). Please clarify.

Individual values instead of bar graphs should be used whenever possible.

It is speculated that OxPhos complexes are improperly assembled after the loss of M1BP. However, there is no data specifically supporting this statement. In fact, the BN-PAGE shows that the molecular weights of each complex appear to be similar between KD and WT muscles which argues against improper assembly. BN-PAGE complexes should be cut out and either run on a mass spec or an SDS gel to assess composition of each complex. Garcia et al. Cell Reports, 2017 provides several additional examples of how to assess complex assembly in Drosophila flight muscles.

Minor Comments

Lines 69-70: This statement implies that mitochondria are not in close contact with the myofibrils in leg muscles. Do you have any evidence to support this statement? In mammalian muscle, oxidative mitochondria (branching mito networks as in fly legs) are in similar contact to the sarcomere compared to cardiac mitochondria (IFM mito network structure) (Figure 4i in Bleck et al, 2018).

Supplemental Figure 1A bottom: It is very difficult to discern between the 8 different "other" mitochondrial transcripts in this figure. A zoomed in inset from 48h APF to adult would be helpful.

Figure 1C: Images are shown from 34h APF but no corresponding data are shown. Text says there was difficulty obtaining clear images from 20-30h APF, but these images are outside that window.

Lines 150-1: It is stated that mitochondria are intercalated between myofibrils which implies that mitochondria are inserted between the contractile apparatus. However, as shown here, mitochondria are present before the myofibrils form. Does this mean the myofibrils are intercalated between mitochondria instead?

Line 207: As written, this suggests mito genes were excluded from analysis. Please clarify.

Reviewer #2 (Remarks to the Author):

The authors have reaffirmed through transcriptional profiling and electron microscopy that mitochondria structure is dynamically altered during muscle formation in Drosophila. Furthermore, the authors suggest that M1BP is a transcription factor regulating this dynamic alteration in mitochondria. Without M1BP, developing fly muscles do not form mitochondria properly, which can impact flight muscle function, but perhaps most strikingly lead to mitochondrial inclusions.

Overall this research is well done and well written, this reviewer appreciates the transparency of providing tables with the RNASeq data to examine. The electron microscopy work is particularly impressive, especially with regards to the rather unique finding of mitochondrial IMM inclusion, and this reviewer believes perhaps that novelty should be emphasized a bit more. I do have some concerns that I would like addressed by authors listed below:

1. Figure 1A/B - Given the different methodologies described to extract RNA from muscle cells at the different stages of development (especially myoblast vs pupal vs adult being 3 different methods), I question how reasonable it may be to compare these to each other in such a way? The pupal stages

would be fair to consider as comparable given the same methodology was used. Similarly, I could not find information describing normalization of the reads; is it transcript reads normalized by transcript size and total reads? Are the total reads drastically different between the myoblast, pupal and adult stages?

2. Whilst the logic for pursuing M1BP as a potential regulator of mitochondrial related transcripts is described, I feel this could be expanded on otherwise the reason for testing this gene seems a little weak. I also believe it would be useful to demonstrate the totality of transcriptional changes for M1BP knockdown rather than just focusing on the mitochondrial transcripts from the beginning.

3. Considering that you have described flight muscles as requiring aerobic glycolysis to maintain flight for periods of time and M1BP RNAi#1 does not completely prevent flight, perhaps it is worth examining the fatiguability of this muscle with compromised aerobic function.

4. The mitochondrial inclusions observed in Figure 3D are particularly striking and I would like to see this emphasized and discussed a little more in the text as to the uniqueness of this phenotype.

5. Whilst figure 4 aims to demonstrate the direct role of M1BP in regulating mitochondrial transcript it succeeds better at demonstrating phenotypically the impact altered mitochondrial transcripts has on mitochondrial function. In order to demonstrate that M1BP is specifically regulating mitochondrial transcripts, chromatin immunoprecipitation (CHIP) of M1BP would be needed to demonstrate the binding (in proximity to differentially expressed mitochondrial genes) and regulation of those genes.

6. The protein quality control induced by this mitochondrial dysfunction is also interesting and it would be particularly interesting to follow up on this by examining whether certain degradation mechanisms are employed to attempt removal of the inclusions. For example, is there any additional ubiquitination present on the mitochondria from M1BP knockdown muscles? Or autophagosome/lysosome concentration in proximity to the inclusions?

7. In order to suggest that this necessity for M1BP transcriptional regulation is predominantly due to developmental requirements and has minimal effects following development you utilized a later acting ACT88F-GAL4 driver which had minimal effects on mitochondria. You reasoned that given similar levels of knockdown compared to MEF2-GAL4 using fluorescent imaging that these reductions are similar and therefore only the timing of action is different. Firstly, I would question the the nature of using fluorescence microscopy as a quantitative measure of knockdown especially when comparing across two different states such as that, why is it not possible to perform qPCR for this?. Secondly, in my experience different GAL4s usually drive different degrees of knockdown efficiency with the same RNAi and can have different phenotypes not just due to the timing. I believe it would therefore be prudent to test this in other ways, for example using the MEF2-GAL40+GAL80TS method you describe, activate this after development and examine whether mitochondrial defects are observable. There are also other late acting muscle drivers such as the MHC-F3.580-GAL4 (<https://bdsc.indiana.edu/Home/Search?presearch=38464>), which in my experience has stronger knockdown efficiency than the ACT88F-GAL4 that may be worth trying in conjunction with M1BP RNAi also to determine whether mitochondrial phenotypes are observable in adults. Adding this would provide better confirmation of whether or not M1BP also regulates mitochondrial transcription following development in addition to during muscle development.

Minor points:

Line 127 (and a few other places)- using the term "half" would generally be preferable to "twice lower"

Line 593 (and other places) - "ON" I presume is overnight but is not defined anywhere

Overall this body of research is quite nicely done, but could benefit from the above considerations. If

the authors are able to address some of these points either through additional experimental work, modifications to the text, or providing sound reasoning for their logic I would strongly support the publication of this research.

Reviewer #3 (Remarks to the Author):

The manuscript titled "M1BP is an essential 1 transcriptional activator of oxidative metabolism during *Drosophila* flight muscle development" examines the involvement of M1BP in mitochondrial biogenesis especially cristae formation and concomitant mitochondrial oxidative function during IFM development in *Drosophila*. It is an interesting piece of study that looks at cristae formation and development during skeletal muscle development. Apart from the novel findings, I am also particularly impressed with the technical skills applied in this study. The scientists have FACS isolated myoblasts and dissected pupal muscles corresponding to the early puparium stage and 64h pupal stage to assess mitochondrial activity. I favor this study. There are certain concerns that when taken care of will make the results more convincing as listed below:

1. A rescue experiment is required where M1BP expression in the knock-down background would reverse the transcriptional changes induced by M1BP RNAi. It is required to show that the downstream effects of M1BP are indeed a result of M1BP depletion. Also, Mef2-Gal4 can drive expression in a subset of neurons [1], so the use of another driver should be considered. Alternatively, express RNAi lines in neurons to identify whether there are any muscle/mitochondrial defects.
2. Line 339: The authors stated, "Altogether these data demonstrate that M1BP either acts alone or in connection with other transcription factors that remain to be identified, in upregulating OXPHOS genes required for complex assembly and integrity during flight muscle development". Genetic interaction studies with crebB, Mef2, hcf, and ewg would present evidence. If it is beyond the scope of this study, the authors can discuss the possibility of interactions.
3. Because mitochondrial and muscle growth take place simultaneously during the IFM development, does muscle development (myofibrillar or muscle fiber development) get affected? It would be nice to see a time-dependent change (especially around 64h APF) in IFM growth in Mef2>M1BP RNAi flies compared to the control flies. It is [2] previously shown that mitochondrial fusion defects during IFM development results in the loss of myofibrillar growth. Since the defects in cristae number Mef2>M1BP RNAi is evident around 64h APF, a stage when the myofibrillar growth rate is at its peak, how does M1BP RNAi affect muscle growth? Considering that the myoblast numbers didn't change (Fig. S2), I would assume that the number of myofibers will not change.
4. Since authors have carefully examined the cristae formation during pupal IFM development, does it mean that M1BP is not required for cristae formation early during the pupal development (say 12-50h APF)? Or M1BP is required for the preservation of cristae post 50h APF in an M1BP-dependent manner? Is it known if the cristae undergo remodeling around 50h APF where early pupal OXPHOS proteins are replaced by newly synthesized OXPHOS as evidenced by a robust increase in their transcription?
5. The authors show in Fig. 3 that mitochondria contain electron-dense inclusions encircled by cristae. An apoptosis assay (TUNEL) might provide additional evidence of the status of mitochondrial viability and function.
6. Results pertaining to late pupal or adult knock-down of M1BP RNAi need corroboration. Act88F-Gal4 is known to express as early as Act88F expression (around 36h APF) begins in the developing IFM. Additionally, the knockdown achieved (26%) is less compared to Mef2>M1BP RNAi (60%). So, the lack of mitochondrial or muscle defects could as well be due to insufficient knock-down! My suggestion would be to include a gene switch Act88F Gal4 to drive knock-down post 64h APF until the adult stage. Similarly, gene switch Mhc-Gal4 can be used to corroborate results of Mef2>M1BP RNAi.
7. Fig. 4C: Is it possible to normalize enzymatic activities to the amount of respective OXPHOS complexes? The graphs in Fig. C otherwise are misleading.
8. The authors show Hsp70 response is localized to the mitochondria. Additionally, according to the authors' data, in addition to mitochondrial Hsp22, cytosolic HSPs also increase in expression (Fig. 5A).

Mitochondrial stress or compensatory pathways are known to elicit cellular proteostasis [3]. Further experiments with muscle extracts and/or mitochondrial extracts for unfolded protein response are required. I can suggest an anti-polyubiquitin (and others) probe for the assessment of ubiquitinated proteins (indicative of misfolded protein aggregates) in insoluble protein fractions of either muscle and/or mitochondrial extracts [4]. These results (also point 3) are important to obtain and discuss as they may support weakened muscles as a cause for pharate lethality and flight defects in the escaper RNAi flies.

9. The definition of "mito-gene" is inconsistent. Line 49 defines them as nuclear-encoded mitochondrial genes. Line 207 defines mito-gene as mitochondrial-encoded genes. Please clarify them in the text and the data in figures/supplementary figure and their corresponding legends.

10. Discussion section: should be modified to discuss the results obtained above. A few sentences on mitochondrial proteases should be discussed as they are a pivotal part of unfolded protein response and related diseases [5].

References:

1. Blanchard, F.J., et al., The transcription factor Mef2 is required for normal circadian behavior in *Drosophila*. *J Neurosci*, 2010. 30(17): p. 5855-65.
2. Rai, M. and U. Nongthomba, Effect of myonuclear number and mitochondrial fusion on *Drosophila* indirect flight muscle organization and size. *Exp Cell Res*, 2013. 319(17): p. 2566-77.
3. Ruan, L., et al., Cytosolic proteostasis through importing of misfolded proteins into mitochondria. *Nature*, 2017. 543(7645): p. 443-446.
4. Rai, M., et al., Analysis of proteostasis during aging with western blot of detergent-soluble and insoluble protein fractions. *STAR Protoc*, 2021. 2(3): p. 100628.
5. Gomez-Fabra Gala, M. and F.N. Vogtle, Mitochondrial proteases in human diseases. *FEBS Lett*, 2021. 595(8): p. 1205-1222.

Reviewer #1 (Remarks to the Author):

The authors investigate the role of M1BP in development of flight muscle mitochondrial structure and composition demonstrating that loss of M1BP leads to reduced cristae, OxPhos complexes, abnormally shaped mitochondria, up regulation of the protein quality control response, and a flightless phenotype. Developmental regulation of mitochondrial ultrastructure in muscles is an area of great interest. However, there are concerns with the generalizability of these flight muscle data as well as several areas that should be better clarified.

We thank the reviewer for their appreciation of the topic of our article and for their important suggestions and corrections, which we have taken on board and detail below.

1. It is unclear how applicable these results are beyond *Drosophila* IFMs. IFMs are asynchronous, have very little sarcoplasmic reticulum, and have individual myofibrils which run the length of the cell, all of which are unlike any known mammalian muscles or the majority of other muscles within the fly. Does the role of M1BP extend to any other muscle types? Assessment of some type of tubular muscle (e.g. leg, TDT) should minimally be performed in order to generalize these results. Additionally, discussion of the mammalian ortholog(s) of M1BP and their known role in muscle development should be addressed.

We thank the reviewer for this comment. Light microscopy is not sufficient to assess whether there is a phenotype in tubular muscles upon M1BP KD, due to the much smaller size of mitochondria (quantified in Avellaneda et al., 2021). To overcome this, we thus used TEM to look at the leg muscle mitochondria and observed that in the smaller tubular muscle mitochondria the amorphous inclusions can also be observed. We performed a leg muscle functional assay (negative geotaxis) and observe that M1BP KD indeed leads to impaired leg muscle function where flies have great difficulty in climbing (94% of adult males stay on the bottom of the tube and 6% in the middle) and their crawling speed is reduced by 90%.

To further assess the generalisability of our findings we looked in another highly metabolically active tissue of the fly, the larval fat body, which is the equivalent of vertebrate liver and adipose tissue. Downregulating M1BP with a fat body-specific driver *cg-GAL4* led to very similar mitochondrial phenotypes than those observed in muscles, including large amorphous inclusions, demonstrating that M1BP function in mitochondria extends to other highly metabolic *Drosophila* tissues.

We have added these leg muscle and fat body data to the new Supplementary Figure 3.

Concerning a M1BP mammalian ortholog, we have previously shown that amongst the 23 ZKSCAN protein members orthologous to M1BP, in the context of autophagy control, it is the vertebrate transcription factor ZKSCAN3 that is a functional homologue of M1BP, where the expression of ZKSCAN3 can prevent premature autophagy induction in the fat body caused by M1BP KD and vice versa, M1BP expression can prevent autophagy in vertebrate cells caused by the nuclear translocation of ZKSCAN3 to the cytoplasm (Barthez et al., 2020, <https://doi.org/10.1038/s41598-020-66377-z>). Even though ZKSCAN3 is a repressor of autophagy in mammals, it was not linked with the control of oxidative metabolism nor muscle development. We nevertheless tested whether ZKSCAN3 overexpression could restore mitochondrial phenotypes triggered by M1BP KD in the flight muscle. We observed that this is not the case, and ZKSCAN3 cannot substitute for M1BP in OXPHOS regulation in flight muscles (see attached figure). We thus do not know the vertebrate homologue of *Drosophila* M1BP in its function in transcriptionally regulating oxidative metabolism in the fly

and we hope that the reviewer agrees that testing the remaining ZKSCAN family members is outside the scope of this manuscript.

2. Does M1BP overexpression result in the opposite effect with increased OxPhos complexes and cristae?

We possess two UAS lines allowing to overexpress M1BP. We initially used the 3xHA-tagged line obtained from FlyORF (FlyORF line #000001) but after numerous highly time-consuming experiments we found that this line in fact produced highly truncated non-functional protein, where more than half of the protein is missing – the line has been withdrawn from the FlyORF resource. The second UAS line, tagged GFP, when crossed with *Mef2*-GAL4 leads to early pupal lethality even at very low temperature of 18°C where GAL4 is little active. Knowing its role in the control of other “housekeeping genes” linked to cellular metabolism (Li and Gilmour PMID 23708796) and ribosomal genes (Baumann and Gilmour PMID 28977400), we thus believe that overexpression of M1BP in the tissues where *Mef2*-Gal4 is active, even at low amounts, is too detrimental to development to assess increased oxidative capacity.

3. Do adenine nucleotide translocase (ANT) and the Pi transporter, both key OxPhos components, follow a similar expression pattern as Complexes I-V?

We thank the reviewer for this point that was missing in our analyses. In *Drosophila* there are two ANT homologues, ANT1 (in *Drosophila* called stress-sensitive B (*sesB*)) and ANT2 that are transcribed from the same promoter (<https://doi.org/10.1093/genetics/153.2.891>). The expression of ANT1 indeed follows the same expression dynamics as OXPHOS complexes, which is not the case for ANT2. We thus believe that ANT1 is the predominant isoform during flight muscle development, consistent with the above mentioned report stating that ANT1 is the predominant isoform in the adult animal.

Concerning the Pi carrier, we found two homologous of the vertebrate Phosphate Carrier Protein, encoded by the *SLC25A3* gene, called Mitochondrial phosphate carrier protein 1 and 2 (*Mpcp1* and 2). In a similar manner to ANTs, *Mpcp1* seems to be the predominant form

during flight myogenesis, following the same expression dynamics as OXPHOS complexes and Ant1.

To extend upon this point, we also looked at the expression of phosphocreatine kinases (in *Drosophila*, arginine kinases), where the predominant form during the IFM development is Argk1 (arginine kinase 1, orthologue of human CKM).

We have included the expression of these genes in the Supplementary Figure 5.

4. Intro, third paragraph: It is well known that both fast and slow-twitch muscles can be oxidative and can have similar mitochondrial content, individual mitochondrial morphology, and mitochondrial network morphology. Discussion of muscle fiber type should discriminate between metabolic and contractile fiber types. Moreover, the short disconnected appearance of glycolytic muscle mitochondria is largely due to 2D imaging in the longitudinal axis. 3D imaging or 2D imaging along the lateral axis shows that glycolytic mitochondria in mammals are also elongated and connected along the lateral axis (e.g. Ogata and Yamasaki 1985, Picard et al 2013, Bleck et al. 2018).

We thank the reviewer for this correction, we modified the introduction paragraph accordingly (lines 54-64).

5. Figure 1C: As stated in the manuscript, development of mitochondrial ultrastructure is critical to mitochondrial function. However, only cristae number is quantified despite the high quality of the majority of the available images. Mitochondrial shape is alluded to several times but should be backed by quantitative assessment. Mitochondrial shape (circularity, aspect ratio, etc), area, perimeter, surface area to volume, length, and cristae width are all informative measures of mitochondrial structural capacity that can be assessed in the images as shown. Mitochondrial content would also be informative and can be easily assessed in the images in Supplemental Figure 1.

We extended the cristae number quantification by quantifications of mitochondrial circularity, aspect ratio (AR), area and perimeter and included these data in the Supplementary figure 4. In line with the shape descriptions stated in the initial manuscript, we observed a significant increase in AR and perimeter (and thus decrease in circularity) between 24 h and 34 h where mitochondria fuse and become large tubules. On the other hand, between 72 h and the pharate stage mitochondria grow laterally and become again more ellipsoid as shown by increased circularity and decreased AR. We detected a significant increase in the area and perimeter between pharate and adult stage further suggesting that biogenesis continues during eclosion. The quantification of volume is not possible in our case since we performed 2D tissue section imaging. For the same reason, we did not quantify cristae length and width that depend largely on the mitochondrial orientation. We have included the new quantification measurements in the new Supplementary figure 4

6. Lines 162-6: It is shown that mitochondrial cristae biogenesis continue through eclosion and is speculated that biogenesis stops “the moment muscles are solicited for flight.” This should be tested by assessing muscles at a second adult time point.

We extended our TEM imaging of IFM mitochondria to 7 and 30 days of adult development and quantified the number of cristae. We acquired also other samples at 2 days to extend upon the data we had. We found that cristae biogenesis indeed continues after eclosion (and does not stop when muscles are solicited for flight as we incorrectly presumed) and thus

thank the reviewer for pointing this out. We found a significant increase in the number of cristae between 2 and 7 days of adult development, followed by a decrease at 30 days to the values observed at 2 days, likely representing the starting of ageing. We have added these important data to the new Supplementary Figure 4.

7. Figure 1 and Figure 2A,C-E use myoblast/APF/adult time points whereas Figure 2B uses embryo, larva, pupa, adult. Timing descriptors should be consistent across figures to ease comparison of data.

We have corrected the time descriptions used in the previous Figure 2B (new Figure 3A) to be more in line with larval/pupal time descriptions used in the other figures.

8. Figure 2B: Does the lethal M1BP phenotype require the absence of M1BP during the larval stage? It is suggested that the early/mid pupal stage is when M1BP is essential. However, in the current data in Figure 2B M1BP, is always knocked down during the larval stage first. M1BP knockdown should be restricted to the early/mid pupal stage and the phenotype assessed to strengthen the claims made. Later experiments show that knockdown from the mid-pupal stage on (Act88F-Gal4) have no phenotype, so this needs to be clarified.

We have included new Gal80 data in the new Figure 3A where RNAi is induced post larval stages that demonstrate there is no M1BP requirement in larval stages to induce the lethal M1BP phenotype.

9. Figure 2C: It is unclear at what timepoint(s) M1BP was knocked down in these experiments.

M1BP was downregulated during whole myogenesis, using a muscle-specific Mef2-GAL4 driver as stated at the bottom of the figure (left panels Mef2>WT, right panels Mef2>M1BP RNAi). We have modified the text (line 311) and corresponding figure legend (line 1412) to clarify this.

10. Figure 3A: The M1BP driven change in mitochondrial structure independent of changes in sarcomere structure seem to argue against the mechanical feedback model where mitochondria and myofibril biogenesis are coordinated together recently proposed by some of these authors (Avellaneda et al, 2021). Please clarify.

We do not think that our data argue against the model proposed by Avellaneda et al, 2021. Avellaneda et al, 2021 shows that perturbing mitochondrial dynamics impacts muscle development, but only in the case of increased fusion/decreased fission. Upon M1BP KD, mitochondria are smaller. As shown in Avellaneda et al, 2021 increasing mitochondrial fission/decreasing fusion making mitochondria smaller does not seem to impact the muscle growth. To clarify this, we have extended our quantification of mitochondria to include, amongst others, mitochondrial content (mitochondrial area with respect to total area). We detected a significantly lower mitochondrial content upon M1BP KD, with no significant change in the mitochondrial density. Together, this suggests that mitochondria are smaller due to a decrease in their biogenesis rather than an increase in their fission. We include these new data in the new Figure 1C.

11. Individual values instead of bar graphs should be used whenever possible.

With the exception of protein blots quantifications that use only n=2, we have now included individual values in all graphs.

12. It is speculated that OxPhos complexes are improperly assembled after the loss of M1BP. However, there is no data specifically supporting this statement. In fact, the BN-PAGE shows that the molecular weights of each complex appear to be similar between KD and WT muscles which argues against improper assembly. BN-PAGE complexes should be cut out and either run on a mass spec or an SDS gel to assess composition of each complex. Garcia et al. Cell Reports, 2017 provides several additional examples of how to assess complex assembly in Drosophila flight muscles.

By stating that OXPHOS complexes are improperly assembled we do not mean that the composition of individual complexes changes. We speculate that due to the decreased expression of assembly factors and some OXPHOS subunits, less OXPHOS complexes can be properly assembled and inserted in the membrane. This is consistent with our results showing less OXPHOS complexes in the membrane using BN-PAGE. To strengthen our claims, we have performed a western blot on purified mitochondrial proteins and observed a 30% decrease of the quantity of ATP5A protein (Complex V subunit) (Added to new Figure 5). Nonetheless, to avoid this confusion, we have changed the caption of the figure to “M1BP KD leads to reduced mitochondrial respiratory complex assembly” and modified the corresponding paragraph.

Minor Comments

Lines 69-70: This statement implies that mitochondria are not in close contact with the myofibrils in leg muscles. Do you have any evidence to support this statement? In mammalian muscle, oxidative mitochondria (branching mito networks as in fly legs) are in similar contact to the sarcomere compared to cardiac mitochondria (IFM mito network structure) (Figure 4i in Bleck et al, 2018).

We did not intend to state that leg mitochondria are not in close proximity to myofibrils which is not correct, but that they adopt more complex branched shapes in comparison to IFM mitochondria. We rephrased lines 73-77 to “Remarkably, the mitochondrial morphology between both muscle types is strikingly different with flight muscle mitochondria containing very dense cristae and being densely packed in between contractile myofibrils, in contrast to more complex shapes of leg muscle mitochondria that locate at the periphery of myofibrils¹⁷.”

Supplemental Figure 1A bottom: It is very difficult to discern between the 8 different “other” mitochondrial transcripts in this figure. A zoomed in inset from 48h APF to adult would be helpful.

We have added an inset from 48h APF onward in the Supplementary Figure 5 to be able to better discern the other mitochondrial transcripts.

Figure 1C: Images are shown from 34h APF but no corresponding data are shown. Text says there was difficulty obtaining clear images from 20-30h APF, but these image are outside that window.

We have corrected the sentence to read “Quantification of cristae at 34 h APF was not performed as they were poorly defined by TEM.” (lines 231-232).

Lines 150-1: It is stated that mitochondria are intercalated between myofibrils which implies that mitochondria are inserted between the contractile apparatus. However, as shown here, mitochondria are present before the myofibrils form. Does this mean the myofibrils are intercalated between mitochondria instead?

Mitochondria are indeed present before the formation of sarcomeres, since they are present already in the myoblasts and larval templates before the fusion. But it is indeed mitochondria that are intercalated in between myofibrils as shown in Avellaneda et al, 2021.

Line 207: As written, this suggests mito genes were excluded from analysis. Please clarify.

In lines 50-51, we define mito-genes as nuclear encoded mitochondrial genes. To respect this definition, we have excluded from the MitoxPlorer database, genes encoded by the mitochondrial genome. We have separately analysed these genes in Supplementary Figure 6. We have changed the sentence (lines 327-330) to “We first looked at the expression of all nuclear-encoded genes related to mitochondrial function from the curated MitoXplorer database ³⁶ from which we excluded glycolysis-related genes and genes encoded by the mitochondrial genome (Mito-genes; n=1064).”

Reviewer #2 (Remarks to the Author):

The authors have reaffirmed through transcriptional profiling and electron microscopy that mitochondria structure is dynamically altered during muscle formation in *Drosophila*. Furthermore, the authors suggest that M1BP is a transcription factor regulating this dynamic alteration in mitochondria. Without M1BP, developing fly muscles do not form mitochondria properly, which can impact flight muscle function, but perhaps most strikingly lead to mitochondrial inclusions.

Overall this research is well done and well written, this reviewer appreciates the transparency of providing tables with the RNASeq data to examine. The electron microscopy work is particularly impressive, especially with regards to the rather unique finding of mitochondrial IMM inclusion, and this reviewer believes perhaps that novelty should be emphasized a bit more. I do have some concerns that I would like addressed by authors listed below:

We thank the reviewer for their useful comments and appreciation of our transcriptomic and imaging data. We detail below replies to the comments raised, and hope they find that the new additions extend our findings in line with the transparency and high quality imaging that the original submission provided.

1. Figure 1A/B - Given the different methodologies described to extract RNA from muscle cells at the different stages of development (especially myoblast vs pupal vs adult being 3 different methods), I question how reasonable it may be to compare these to each other in such a way? The pupal stages would be fair to consider as comparable given the same methodology was used. Similarly, I could not find information describing normalization of the reads; is it transcript reads normalized by transcript size and total reads? Are the total reads drastically different between the myoblast, pupal and adult stages?

We thank the reviewer for raising this important point regarding the normalisation of the RNAseq reads. The data we graphically present of transcription of nuclear genes with mitochondrial function are taken from Spletter et al *Elife* 2018 – yet since mitochondrial biogenesis was not the scope of that study, the data have been little emphasised to demonstrate the important and large-scale upregulation of expression. In the Spletter et al study, the authors indeed sequenced to much deeper sequencing reads during pupal time points, to take into consideration the massive increases in (particularly) sarcomeric and OXPHOS gene expression at these stages (1000s of times more upregulated than basal gene expression; see Supplementary Figure 5). Due to this rather unique developmental situation, normalisation by the classical RPKM (to compare transcripts within a sample) or TPM (to compare transcripts across different samples) cannot be performed since these methods of normalisation, using sequencing depth as reference, would result in genes whose expression don't change over time, appearing to go down in expression. The authors of those data therefore used the normalisation data within the DESeq2 package, which uses the median of ratios for each gene in a sample, to normalise the gene's read count over its geometric mean across all samples (Anders and Huber *Genome Biol*, 2010, vol. 11(R106)). This allows for global normalisation across all samples being studied, while correctly reporting no differences in gene expression for those genes not subject to the very large amplitudes in gene expression observed during flight muscle biogenesis. We have clarified the method of normalisation used in the Methods section.

As for comparing the different stages from our RNA-seq data using RNA preparations from myoblasts, pupal and adult stages (Supplementary Data 2-5), the data are comparable

since we use only differential gene expression analyses of each stage. Thus the sequencing data are analysed within each stage only and not amongst stages.

2. Whilst the logic for pursuing M1BP as a potential regulator of mitochondrial related transcripts is described, I feel this could be expanded on otherwise the reason for testing this gene seems a little weak. I also believe it would be useful to demonstrate the totality of transcriptional changes for M1BP knockdown rather than just focusing on the mitochondrial transcripts from the beginning.

We added the list of all differentially expressed genes as well as Gene Ontology analysis in the Supplementary Tables 2-5. We modified the order of Figures 1 to 3 (which is also the order of how the experiments were conducted) and rewrote the text correspondingly to better explain our logic. We strongly hope this helps with the overall understanding of the logic of the study.

3. Considering that you have described flight muscles as requiring aerobic glycolysis to maintain flight for periods of time and M1BP RNAi#1 does not completely prevent flight, perhaps it is worth examining the fatiguability of this muscle with compromised aerobic function.

We performed fatigability test with M1BP RNAi #1. We tapped the flies three times, let them recover a few seconds and repeated three times to try to deplete the available glucose and then subjected them to the flight test. In this condition, we found 14% of WT flies landing on the bottom (12% more compared to the classical flight test). For M1BP RNAi#1 we found 72% of flies in the bottom (representing 27% more than in the classical flight test). Thus, IFM indeed display higher fatiguability in this condition. We thank the reviewer for suggesting this point and present these data in Supplementary Figure 2.

4. The mitochondrial inclusions observed in Figure 3D are particularly striking and I would like to see this emphasized and discussed a little more in the text as to the uniqueness of this phenotype.

We have modified the order of Figures and the text accordingly to emphasise this phenotype.

5. Whilst figure 4 aims to demonstrate the direct role of M1BP in regulating mitochondrial transcript it succeeds better at demonstrating phenotypically the impact altered mitochondrial transcripts has on mitochondrial function. In order to demonstrate that M1BP is specifically regulating mitochondrial transcripts, chromatin immunoprecipitation (CHIP) of M1BP would be needed to demonstrate the binding (in proximity to differentially expressed mitochondrial genes) and regulation of those genes.

We agree with the reviewer. We extensively tried to perform ChIP-seq of M1BP in the adult IFM. Nevertheless, even with 180 adult hand-dissected flight muscles (which we repeated three times), we were not able to obtain a sufficient amount of chromatin for immunoprecipitation. We have extensive experience in ChIP-seq experiments, but are at a loss as to comprehend our inability to perform this on adult flight muscle – perhaps the hyper-stretched adult muscle prevents nuclear release required for proper chromatin extraction? We are bitterly disappointed by not being able to perform this and we hope the reviewer appreciates the efforts made.

Despite the failings in a ChIP-seq analysis, and to allay the reviewer's concerns, we performed an extensive *in silico* analysis of possible M1BP targets and partners as also suggested by the reviewer #3. The M1BP binding motif is very robust and thus finding it very significantly present at the promoters of many OXPHOS genes is highly indicative of M1BP binding. Through such an analysis, we found that the binding motif of M1BP is the most significantly enriched motif in the promoters of OXPHOS-genes. We extended this analysis to other known OXPHOS regulators mentioned in the first version of the manuscript and performed a co-occurrence analysis. We included these data in the new Figure 4. While these analyses do not substitute for the gold-standard ChIP-seq analyses, we hope the reviewer finds that the robust M1BP motif significantly enriched at numerous OXPHOS promoters is suggestive of M1BP binding, and that the new extended analyses to other TF binding motifs adds a new future dimension to potential cooperative transcription factor regulation of OXPHOS gene expression.

6. The protein quality control induced by this mitochondrial dysfunction is also interesting and it would be particularly interesting to follow up on this by examining whether certain degradation mechanisms are employed to attempt removal of the inclusions. For example, is there any additional ubiquitination present on the mitochondria from M1BP knockdown muscles? Or autophagosome/lysosome concentration in proximity to the inclusions?

We performed western blot analyses on purified mitochondria and indeed observed an increased amount of ubiquitinated proteins on mitochondria from M1BP KD muscles. We thank the reviewer for this suggestion, and have included these data in the new Figure 6.

Due to their large size and position in between myofibrils we think that mitochondria are not easily eliminated by the autophagy pathway. We nevertheless observed several mitochondria that seem very electron lucid and permeabilised, probably being eliminated upon M1BP KD, we add such an example in the new Figure 1.

7. In order to suggest that this necessity for M1BP transcriptional regulation is predominantly due to developmental requirements and has minimal effects following development you utilized a later acting ACT88F-GAL4 driver which had minimal effects on mitochondria. You reasoned that given similar levels of knockdown compared to MEF2-GAL4 using fluorescent imaging that these reductions are similar and therefore only the timing of action is different. Firstly, I would question the the nature of using fluorescence microscopy as a quantitative measure of knockdown especially when comparing across two different states such as that, why is it not possible to perform qPCR for this?. Secondly, in my experience different GAL4s usually drive different degrees of knockdown efficiency with the same RNAi and can have different phenotypes not just due to the timing. I believe it would therefore be prudent to test this in other ways, for example using the MEF2-GAL40+GAL80TS method you describe, activate this after development and examine whether mitochondrial defects are observable. There are also other late acting muscle drivers such as the MHC-F3.580-GAL4 (<https://bdsc.indiana.edu/Home/Search?presearch=38464>), which in my experience has stronger knockdown efficiency than the ACT88F-GAL4 that may be worth trying in conjunction with M1BP RNAi also to determine whether mitochondrial phenotypes are observable in adults. Adding this would provide better confirmation of whether or not M1BP also regulates mitochondrial transcription following development in addition to during muscle development.

We have qPCR data from adult IFM with both *Mef2*-GAL4 and *Act88F*-GAL4 driven M1BP KD. In our hand the M1BP KD is similar between the two drivers (decrease of M1BP mRNA levels

in both cases by 60%). We have added the qPCR data for *Mef2*-GAL4 in the Supplementary Figure 1. As suggested, we tested M1BP KD with *Mhc*-GAL4 driver and we did not observe any mitochondrial phenotype. We thus believe that the function of M1BP is predominant during the first half of myogenesis when cristae formation starts and OXPHOS genes start to be upregulated.

We understand the confusions the Act88F data bring, especially in comparison with the GAL80 data: Act88F is at 25°C, Gal80 is at either 18° (RNAi off) or 29° (RNAi on), thus both have very different developmental time frames. The precise timing of M1BP function is not a major point this article wants to convey and is very difficult to assess with negative results (such as the absence of phenotype with *Act88F* and *Mhc* GAL4 drivers). In light of this, we have decided to remove the Act88F data from the manuscript, without any detriment to the data or message conveyed.

Minor points:

Line 127 (and a few other places)- using the term "half" would generally be preferable to "twice lower"

We have corrected the sentence.

Line 593 (and other places) - "ON" I presume is overnight but is not defined anywhere

We defined ON as overnight.

Overall this body of research is quite nicely done, but could benefit from the above considerations. If the authors are able to address some of these points either through additional experimental work, modifications to the text, or providing sound reasoning for their logic I would strongly support the publication of this research.

Reviewer #3 (Remarks to the Author):

The manuscript titled “M1BP is an essential 1 transcriptional activator of oxidative metabolism during *Drosophila* flight muscle development” examines the involvement of M1BP in mitochondrial biogenesis especially cristae formation and concomitant mitochondrial oxidative function during IFM development in *Drosophila*. It is an interesting piece of study that looks at cristae formation and development during skeletal muscle development. Apart from the novel findings, I am also particularly impressed with the technical skills applied in this study. The scientists have FACS isolated myoblasts and dissected pupal muscles corresponding to the early puparium stage and 64h pupal stage to assess mitochondrial activity. I favor this study. There are certain concerns that when taken care of will make the results more convincing as listed below:

We thank the reviewer for their comments and are pleased that they find the article interesting and are impressed by the technical aspects of the techniques used. We address their concerns, outlined below, and hope they find that the substantially revised manuscript alleviates any concerns that they had to the conceivability of our data.

1. A rescue experiment is required where M1BP expression in the knock-down background would reverse the transcriptional changes induced by M1BP RNAi. It is required to show that the downstream effects of M1BP are indeed a result of M1BP depletion. Also, *Mef2-Gal4* can drive expression in a subset of neurons [1], so the use of another driver should be considered. Alternatively, express RNAi lines in neurons to identify whether there are any muscle/mitochondrial defects.

We thank the reviewer for their comment aimed at controlling for the effects seen as being due to M1BP loss of function in flight muscle. We included two independent RNAi lines to negate against any potential off-target phenotypes and have now quantitated all data using these two lines, with identical results. Nevertheless, the reviewer has suggested over-expressing M1BP in an M1BP RNAi condition to rescue the effects seen. It is not possible to derive an RNAi-refractory M1BP expression construct and thus such an experiment would rely on “overwhelming” the RNAi machinery directed against M1BP with “sense M1BP RNA” so that enough is available to produce M1BP protein. While we have doubts that this would be a successful experiment (*Gal4* UAS dilution notwithstanding), we nevertheless attempted these experiments. We possess two UAS lines allowing to overexpress M1BP. We initially used the 3xHA-tagged line obtained from FlyORF (FlyORF line #000001) but after numerous highly time-consuming experiments we found that this line in fact produced highly truncated non-functional protein, where more than half of the protein is missing – the line has been withdrawn from the FlyORF resource. The second UAS line, tagged GFP, when crossed with *Mef2-GAL4* leads to early pupal lethality even at very low temperature of 18°C where GAL4 is little active. Knowing its role in the control of other “housekeeping genes” linked to cellular metabolism (Li and Gilmour PMID 23708796) and ribosomal genes (Baumann and Gilmour PMID 28977400), we thus believe that overexpression of M1BP in tissues where *Mef2-Gal4* is active, even at low amounts, is too detrimental to development to assess increased oxidative capacity. In conclusion, for the reasons described above, we were unable to conduct a successful rescue experiment.

We were nevertheless able to perform several other specificity controls. We expressed M1BP RNAi (#1 and #2) in the neurons using *elav-GAL4* driver and did not observe any muscle phenotype, meaning that the phenotypes we observed with the *Mef2-GAL4* driver do not come from M1BP expression in neurons (even at 29°C). To expand upon this point, knowing that *Mef2* is also expressed in the fat body, we used a *cg-GAL4* driver and similarly to *elav-GAL4* did not find any muscle phenotype. Using *cg-Gal4*-driven M1BP RNAi, we did however observe

the same mitochondrial inclusions in fat body cells, which we include in new Supplementary Figure 3C.

Finally, we confirmed our findings (muscle paralysis and mitochondrial inclusions) with another muscle-specific GAL4 driver, *Him*-GAL4 that in our hands downregulates M1BP expression until the adult stage with a similar strength as the *Mef2*-GAL4 driver. We have added these data in the new Supplementary Figures 2 and 3.

2. Line 339: The authors stated, “Altogether these data demonstrate that M1BP either acts alone or in connection with other transcription factors that remain to be identified, in upregulating OXPHOS genes required for complex assembly and integrity during flight muscle development”. Genetic interaction studies with *crebB*, *Mef2*, *hcf*, and *ewg* would present evidence. If it is beyond the scope of this study, the authors can discuss the possibility of interactions.

Unfortunately, there are no genetic interaction data available for M1BP and performing them are beyond the scope of the study. Yet, to assess the possible genetic networks controlling OXPHOS in the IFM we extended the in silico analyses of possible M1BP targets and partners. Taking advantage of the strong DNA binding motif of M1BP we checked each OXPHOS gene encoding structural subunits and assembly factors for the presence or absence of this motif in the promoter region. We found that the DNA binding motif of M1BP is the most significantly enriched motif in the promoters of OXPHOS-genes. We extended this analysis to other known OXPHOS regulators mentioned in the first version of the manuscript and performed a co-occurrence analysis. We have included these data in the new Figure 4.

3. Because mitochondrial and muscle growth take place simultaneously during the IFM development, does muscle development (myofibrillar or muscle fiber development) get affected? It would be nice to see a time-dependent change (especially around 64h APF) in IFM growth in *Mef2*>*M1BP* RNAi flies compared to the control flies. It is [2] previously shown that mitochondrial fusion defects during IFM development results in the loss of myofibrillar growth. Since the defects in cristae number *Mef2*>*M1BP* RNAi is evident around 64h APF, a stage when the myofibrillar growth rate is at its peak, how does M1BP RNAi affect muscle growth? Considering that the myoblast numbers didn't change (Fig. S2), I would assume that the number of myofibers will not change.

We did not observe any changes in myofibril development upon M1BP KD as stated in the initial version of the manuscript. To verify this quantitatively we quantified the width of myofibrils as well as the sarcomere length and we observed no significant difference. We also included confocal sections of whole hemi-thoraces for better visualisation. We include these data in Figure 1.

4. Since authors have carefully examined the cristae formation during pupal IFM development, does it mean that M1BP is not required for cristae formation early during the pupal development (say 12-50h APF)? Or M1BP is required for the preservation of cristae post 50h APF in an M1BP-dependent manner? Is it known if the cristae undergo remodeling around 50h APF where early pupal OXPHOS proteins are replaced by newly synthesized OXPHOS as evidenced by a robust increase in their transcription?

We thank the reviewer for this interesting point. We do not believe that M1BP is involved in the regulation of cristae formation before 50h. In our quantifications of cristae, we did not find any difference in the number of cristae between the myoblast and 50h APF stages, suggesting

there is no cristae formation in absolute numbers (we cannot say about the dynamic remodelling in vivo that we did not study). We demonstrated that the cristae formation/biogenesis starts after 50h APF where we believe M1BP is required.

We are not aware of any studies looking of cristae remodelling at 50h APF, we do not know whether OXPHOS proteins are replaced, although there is *de novo* OXPHOS proteins added as well as new cristae.

5. The authors show in Fig. 3 that mitochondria contain electron-dense inclusions encircled by cristae. An apoptosis assay (TUNEL) might provide additional evidence of the status of mitochondrial viability and function.

To our knowledge the TUNEL assay is not applicable to indirect flight muscles. We nevertheless did observe several instances of dying mitochondria in our electron microscopy samples, as highlighted by highly electron lucid mitochondria. We have added such an example in Figure 1.

6. Results pertaining to late pupal or adult knock-down of M1BP RNAi need corroboration. Act88F-Gal4 is known to express as early as Act88F expression (around 36h APF) begins in the developing IFM. Additionally, the knockdown achieved (26%) is less compared to Mef2>M1BP RNAi (60%). So, the lack of mitochondrial or muscle defects could as well be due to insufficient knock-down! My suggestion would be to include a gene switch Act88F Gal4 to drive knock-down post 64h APF until the adult stage. Similarly, gene switch Mhc-Gal4 can be used to corroborate results of Mef2>M1BP RNAi.

As suggested also by the reviewer 2, we downregulated M1BP with *Mhc*-GAL4 and did not observe any mitochondrial phenotype like in the case of *Act88F*-GAL4. We thus believe that the function of M1BP is predominant at the first half of myogenesis when cristae biogenesis starts and OXPHOS genes start to be upregulated. To our knowledge, the Gene-Switch in the pupa is not feasible since pupae cannot be fed during this stage to activate the GS system. Knowing the confusion the Act88F data bring (difficulty in interpreting the absence of effects) and its dispensability (as discussed in response to point 7 of Reviewer 2) we preferred to remove these data from the manuscript, which does not detract from the message the manuscript delivers.

7. Fig. 4C: Is it possible to normalize enzymatic activities to the amount of respective OXPHOS complexes? The graphs in Fig. C otherwise are misleading.

We thank the reviewer for this point and apologise that the original representations could be found misleading. We have thus normalised the amounts of complexes in the new Figure 5 as a “percent relative to WT” and did the same for the enzymatic activities to be directly comparable.

8. The authors show Hsp70 response is localized to the mitochondria. Additionally, according to the authors’ data, in addition to mitochondrial Hsp22, cytosolic HSPs also increase in expression (Fig. 5A). Mitochondrial stress or compensatory pathways are known to elicit cellular proteostasis [3]. Further experiments with muscle extracts and/or mitochondrial extracts for unfolded protein response are required. I can suggest an anti-polyubiquitin (and others) probe for the assessment of ubiquitinated proteins (indicative of misfolded protein aggregates) in insoluble protein fractions of either muscle and/or mitochondrial extracts [4]. These results (also point 3) are important to

obtain and discuss as they may support weakened muscles as a cause for pharate lethality and flight defects in the escaper RNAi flies.

We thank the reviewer for these very helpful suggestions. We have confirmed the increase of Hsp70 in the mitochondrial protein fraction by western blots. As also suggested by the reviewer #2, we observed an increase in the quantity of polyubiquitinated proteins on the surface of mitochondria upon M1BP KD. We further fractionated the mitochondria into soluble and insoluble fractions and found the presence of the complex V subunit in the insoluble fraction upon M1BP KD and not in the WT, further suggesting OXPHOS proteins change their solubility upon M1BP KD (added to the new Figure 5).

We have additionally extracted soluble and insoluble fractions from the whole muscle for the polyubiquitin analyses and found several instances of proteins that were lowly or not polyubiquitinated in the WT compared to M1BP KD and of proteins that have been already polyubiquitinated in the WT but whose quantity increased in the M1BP KD, demonstrating M1BP KD increases ubiquitination of proteins. We have included these data in the new Figure 6.

9. The definition of “mito-gene” is inconsistent. Line 49 defines them as nuclear-encoded mitochondrial genes. Line 207 defines mito-gene as mitochondrial-encoded genes. Please clarify them in the text and the data in figures/supplementary figure and their corresponding legends.

In lines 50-51, we define mito-genes as nuclear encoded mitochondrial genes. To respect this definition, we have excluded from the MitoXplorer database genes encoded by the mitochondrial genome. We have separately analysed these genes in the Supplementary Figure 4. We changed the sentence (lines 327-330) to “We first looked at the expression of all nuclear-encoded genes related to mitochondrial function from the curated MitoXplorer database ³⁶ from which we excluded glycolysis-related genes and genes encoded by the mitochondrial genome (Mito-genes; n=1064).”

10. Discussion section: should be modified to discuss the results obtained above. A few sentences on mitochondrial proteases should be discussed as they are a pivotal part of unfolded protein response and related diseases [5].

We have modified the discussion section accordingly, and also provide an addition paragraph discussing mitochondrial proteases (lines 644-657).

References:

1. Blanchard, F.J., et al., The transcription factor Mef2 is required for normal circadian behavior in *Drosophila*. *J Neurosci*, 2010. 30(17): p. 5855-65.
2. Rai, M. and U. Nongthomba, Effect of myonuclear number and mitochondrial fusion on *Drosophila* indirect flight muscle organization and size. *Exp Cell Res*, 2013. 319(17): p. 2566-77.
3. Ruan, L., et al., Cytosolic proteostasis through importing of misfolded proteins into mitochondria. *Nature*, 2017. 543(7645): p. 443-446.
4. Rai, M., et al., Analysis of proteostasis during aging with western blot of detergent-soluble and insoluble protein fractions. *STAR Protoc*, 2021. 2(3): p. 100628.
5. Gomez-Fabra Gala, M. and F.N. Vogtle, Mitochondrial proteases in human diseases. *FEBS Lett*, 2021. 595(8): p. 1205-1222.

REVIEWERS' COMMENTS

Reviewer #1 (Remarks to the Author):

The authors have done a very thorough job responding to all the reviewer comments, and I have no remaining concerns about this manuscript.

Reviewer #2 (Remarks to the Author):

The authors have worked hard to address my concerns, and have substantially improved the manuscript.

Although CHIP-Seq data would be ideal, I understand this is not always possible but the authors have included data that helps to establish M1BP as a direct regulator of nuclear encoded mitochondrial target genes suggested.

Overall, the authors have done an excellent job and I thank them for the thought and effort that has gone into the revisions. I support the publication of this revised manuscript.

Reviewer #1 (Remarks to the Author):

The authors have done a very thorough job responding to all the reviewer comments, and I have no remaining concerns about this manuscript.

We thank the reviewer for taking the time to re-review our manuscript and are thoroughly delighted that we have responded to all the concerns they had on the original submission.

Reviewer #2 (Remarks to the Author):

The authors have worked hard to address my concerns, and have substantially improved the manuscript.

Although CHIP-Seq data would be ideal, I understand this is not always possible but the authors have included data that helps to establish M1BP as a direct regulator of nuclear encoded mitochondrial target genes suggested.

Overall, the authors have done an excellent job and I thank them for the thought and effort that has gone into the revisions. I support the publication of this revised manuscript.

We thank the reviewer for taking the time to re-review our manuscript and are thoroughly delighted that they now support publication of the revised manuscript.